EMBO
Molecular Medicine

# Coupling of mitochondrial function and skeletal muscle fiber type by a miR-499/Fnip1/AMPK circuit

Jing Liu[1,†], Xijun Liang[1,†], Danxia Zhou[1], Ling Lai[2], Liwei Xiao[1], Lin Liu[1], Tingting Fu[1], Yan Kong[3], Qian Zhou[1], Rick B Vega[2], Min-Sheng Zhu[1], Daniel P Kelly[2], Xiang Gao[1] & Zhenji Gan[1,*]

## Abstract

Upon adaption of skeletal muscle to physiological and pathophysiological stimuli, muscle fiber type and mitochondrial function are coordinately regulated. Recent studies have identified pathways involved in control of contractile proteins of oxidative-type fibers. However, the mechanism for coupling of mitochondrial function to the muscle contractile machinery during fiber type transition remains unknown. Here, we show that the expression of the genes encoding type I myosins, *Myh7/Myh7b* and their intronic miR-208b/miR-499, parallels mitochondrial function during fiber type transitions. Using *in vivo* approaches in mice, we found that miR-499 drives a PGC-1α-dependent mitochondrial oxidative metabolism program to match shifts in slow-twitch muscle fiber composition. Mechanistically, miR-499 directly targets *Fnip1*, an AMP-activated protein kinase (AMPK)-interacting protein that negatively regulates AMPK, a known activator of PGC-1α. Inhibition of *Fnip1* reactivated AMPK/PGC-1α signaling and mitochondrial function in myocytes. Restoration of the expression of miR-499 in the mdx mouse model of Duchenne muscular dystrophy (DMD) reduced the severity of DMD. Thus, we have identified a miR-499/Fnip1/AMPK circuit that can serve as a mechanism to couple muscle fiber type and mitochondrial function.

**Keywords** contractile fiber type; gene regulation; microRNA; mitochondrial function; muscle
**Subject Categories** Metabolism; Musculoskeletal System

## Introduction

Contractile fiber type and energy production are two key determinants of skeletal muscle function. Skeletal muscle fibers exhibit significant physiological plasticity, reflected in different patterns of coordinated metabolic and structural gene expression (Pette & Staron, 2000; Zierath & Hawley, 2004; Schiaffino & Reggiani, 2011). Muscle fibers are classified into slow-twitch (type I) and fast-twitch (type II) based on the myosin heavy chain (MHC) gene expressed (Pette & Staron, 2000; Zierath & Hawley, 2004; Schiaffino & Reggiani, 2011; Liu *et al*, 2015). Type I myofibers are high endurance and are mitochondria-rich (red), relying largely on mitochondrial oxidative metabolism for ATP production (Pette & Staron, 2000; Zierath & Hawley, 2004; Schiaffino & Reggiani, 2011; Liu *et al*, 2015). In contrast, type II myofibers are low endurance and contain fewer mitochondria, and primarily rely on glycolytic burst metabolism for energy production (Pette & Staron, 2000; Zierath & Hawley, 2004; Schiaffino & Reggiani, 2011; Liu *et al*, 2015).

During development, and upon adaption to physiological stimuli and systemic disease, skeletal muscle fibers undergo extensive metabolic and structural remodeling (Holloszy & Coyle, 1984; Booth & Thomason, 1991; Baldwin & Haddad, 2001; Bassel-Duby & Olson, 2006; Rowe *et al*, 2014; Neufer *et al*, 2015). During fiber type transition, the contractile machinery and energy production system must be precisely coordinated to maintain muscle function (Schiaffino & Reggiani, 2011). Exercise training is effective in improving muscle fitness by increasing the proportion of slow-twitch muscle fibers and by promoting mitochondrial oxidative capacity (Holloszy & Coyle, 1984; Booth & Thomason, 1991; Baldwin & Haddad, 2001; Bassel-Duby & Olson, 2006; Rowe *et al*, 2014; Neufer *et al*, 2015). Conversely, skeletal muscle dysfunction, including reduced numbers of slow-twitch muscle fibers and diminished fuel-burning capacity, is linked to several human diseases, including metabolic disorders, heart failure, and muscular dystrophy (Holloszy & Coyle, 1984; Booth & Thomason, 1991; Baldwin & Haddad, 2001; Bassel-Duby & Olson, 2006; Neufer *et al*, 2015).

The mechanism for precise coupling of mitochondrial function and muscle contractile machinery during fiber type transition remains unknown. Previous studies using genetically modified mice have provided important clues. The myosin heavy chain genes, *Myh7/Myh7b*, have been defined as key regulators of muscle fiber type switching by encoding a network of intronic miRNAs (van Rooij *et al*, 2009; Bell *et al*, 2010). Specifically, miR-208b, which is encoded within the type I MHC (*Myh7*) gene, and miR-499, which is

1  State Key Laboratory of Pharmaceutical Biotechnology and MOE Key Laboratory of Model Animals for Disease Study, Model Animal Research Center of Nanjing University, Nanjing, China
2  Diabetes and Obesity Research Center, Sanford Burnham Prebys Medical Discovery Institute, Orlando, FL, USA
3  Department of Biochemistry and Molecular Biology, School of Medicine, Southeast University, Nanjing, China
*Corresponding author. Tel: +86 25 58641546; Fax: +86 25 58641500; E-mail: ganzj@nju.edu.cn
†These authors contributed equally to this work

 

encoded within another type I MHC (*Myh7b*) gene, have been shown to drive a slow-twitch fiber program by downregulating transcriptional repressors that suppress a broad program of slow-twitch contractile protein gene expression (Hagiwara *et al*, 2005, 2007; Ji *et al*, 2007; van Rooij *et al*, 2009; Bell *et al*, 2010; Quiat *et al*, 2011). We have recently found that nuclear receptor ERRγ directly activates the promoters driving the expression of miR-499 and miR-208b in muscle (Gan *et al*, 2013). In addition, the expression of miR-499 positively correlates with slow-twitch fiber proportion and with metabolic parameters, such as $VO_{2max}$ and $ATP_{max}$, in humans (Gan *et al*, 2013). However, whether and how these miRNAs orchestrate mitochondrial function to match the slow-twitch muscle fiber is unclear.

In this study, we found that the expression of *Myh7b/miR-499* and *Myh7/miR-208b* parallels the mitochondrial respiration capacity during muscle fiber type transition. We speculated that the intronic miR-499 and miR-208b might regulate the correlated changes in mitochondrial function during the transition phase. Using transgenic and loss-of-function mouse lines, and primary skeletal myotubes in culture, we found that a regulatory circuitry comprised of Fnip1/AMPK drives PGC-1α-dependent mitochondrial function downstream of miR-499. Thus, our results suggest a molecular mechanism for the adaptive orchestration of mitochondrial function muscle fiber type.

## Results

### The expression of miR-499 and miR-208b parallels mitochondrial function during muscle fiber type transition

As an initial step, to explore whether miRNAs in myosin genes orchestrate mitochondrial function to match muscle contractile fiber type, we examined the expression patterns of miR-499 and miR-208b during muscle fiber type transition. Consistent with previously published data (van Rooij *et al*, 2009; Bell *et al*, 2010), the induced expression of *Myh7* and *Myh7b* genes paralleled the elevated expression of miR-208b and miR-499 during differentiation of skeletal myoblasts into mature myotubes (Fig 1A). A marked increase in mitochondrial respiration capacity correlated with upregulation of miR-499 and miR-208b during the transition phase (Fig 1A and B). Similar results were obtained when we compared different muscle types from adult wild-type mice. As shown in Fig 1C, miR-499 and miR-208b were expressed preferentially in slow-fiber-dominant soleus muscle. Isolated mitochondrial respiration studies demonstrated that pyruvate-driven mitochondrial respiration rates were significantly higher in soleus (Sol) muscle compared to fast-fiber-enriched white vastus (WV) muscle (Fig 1D). Together, these results showed dynamic expression of miR-499 and miR-208b that paralleled mitochondrial function during muscle fiber type transition.

### miR-499 drives a mitochondrial oxidative metabolism program in skeletal muscle

To investigate the potential role of miR-499 and miR-208b in regulating mitochondrial function, we analyzed the muscle fuel utilization of transgenic mice overexpressing miR-499 in muscle

(MCK-miR-499) given that miR-499 and miR-208b function redundantly *in vivo* (van Rooij *et al*, 2009; Bell *et al*, 2010). The real-time respiratory exchange ratio (RER) was measured during a run-to-exhaustion exercise protocol. The MCK-miR-499 mice had a lower RER than did their NTG littermate controls (Fig 1E), indicative of a tendency to preferentially oxidize fat over carbohydrate, consistent with enhanced exercise performance in these mice. Notably, levels of blood lactate following exercise were significantly lower in both younger and older MCK-miR-499 mice than in their NTG littermate controls (Fig 1F). Mitochondrial respiration rates were determined in the plantaris muscle of the MCK-miR-499 mice and corresponding NTG controls. Consistent with the increased aerobic metabolism in MCK-miR-499 mice, pyruvate-driven state 3 respiration rates were significantly higher in MCK-miR-499 muscle compared to the NTG controls (Fig 1G). Together, these results suggest that MCK-miR-499 muscle has increased mitochondrial function.

Comparative analysis of RNA isolated from gastrocnemius (GC) muscle of the MCK-miR-499 mice and NTG littermate controls, confirmed increased expression of many genes encoding slow contractile proteins in MCK-miR-499 muscle (Appendix Table S1). Notably, gene ontology (GO)-based analysis revealed that the primary upregulated genes were mitochondrial metabolic genes (Fig 2A). A broad array of genes encoding a variety of components of the mitochondria were induced in MCK-miR-499 GC muscle (Appendix Table S2). Real-time PCR confirmed that the expression of nuclear-encoded genes (*Ndufa1, Ndufs4, Uqcr11, Cox7a1, Cox7b, Atp5e,* and *Atp5l*) and mitochondrial-encoded genes (mt-*Nd2*, mt-*Co1*, mt-*Atp6*, and mt-*Cytb*) was induced in the GC muscle of MCK-miR-499 mice compared to NTG controls (Fig 2B). The expression of the oxidative biomarkers myoglobin and cytochrome c was also induced in the MCK-miR-499 GC muscles compared to NTG controls (Fig 2C). Moreover, we also found an increased expression of biomarker genes associated with fatty acid uptake (*Cd36* and *Lpl*) in MCK-miR-499 muscle (Fig 2D). In addition, the expression of the glucose oxidation biomarker *Ldhb* (Gan *et al*, 2011) was increased, whereas *Ldha* gene expression was decreased in MCK-miR-499 muscle (Fig 2D). The expected LDH isoenzyme activity shifts were confirmed by gel-activity studies (Fig 2E and F).

Consistent with the gene expression results, metachromatic myosin I ATPase staining confirmed a dramatic increase in type I muscle fibers in MCK-miR-499 GC muscle (NTG, $126 \pm 42$ per section versus MCK-miR-499, $551 \pm 43$ per section, $n = 5$ mice per group, $P = 0.0006$) (Fig 2G). To further evaluate the effect of miR-499 on muscle oxidative mitochondrial activity, we performed histochemical staining for succinate dehydrogenase (SDH) and α-glycerophosphate dehydrogenase (α-GPDH), which are hallmarks for oxidative and glycolytic metabolism, respectively (Schiaffino & Reggiani, 2011). As expected, the SDH enzymatic activities were higher (NTG, $30.5 \pm 2.0\%$ versus MCK-miR-499, $58.7 \pm 1.3\%$, $n = 4$ mice per group, $P < 0.0001$), whereas the α-GPDH activities were lower in the GC muscle of MCK-miR-499 mice compared to their NTG littermate controls (NTG, $74.8 \pm 1.5\%$ versus MCK-miR-499, $31.6 \pm 1.4\%$, $n = 5$ mice per group, $P < 0.0001$) (Fig 2G). Together, these results demonstrate that miR-499 reprograms muscle for increased mitochondrial oxidative capacity, as well as fiber type.

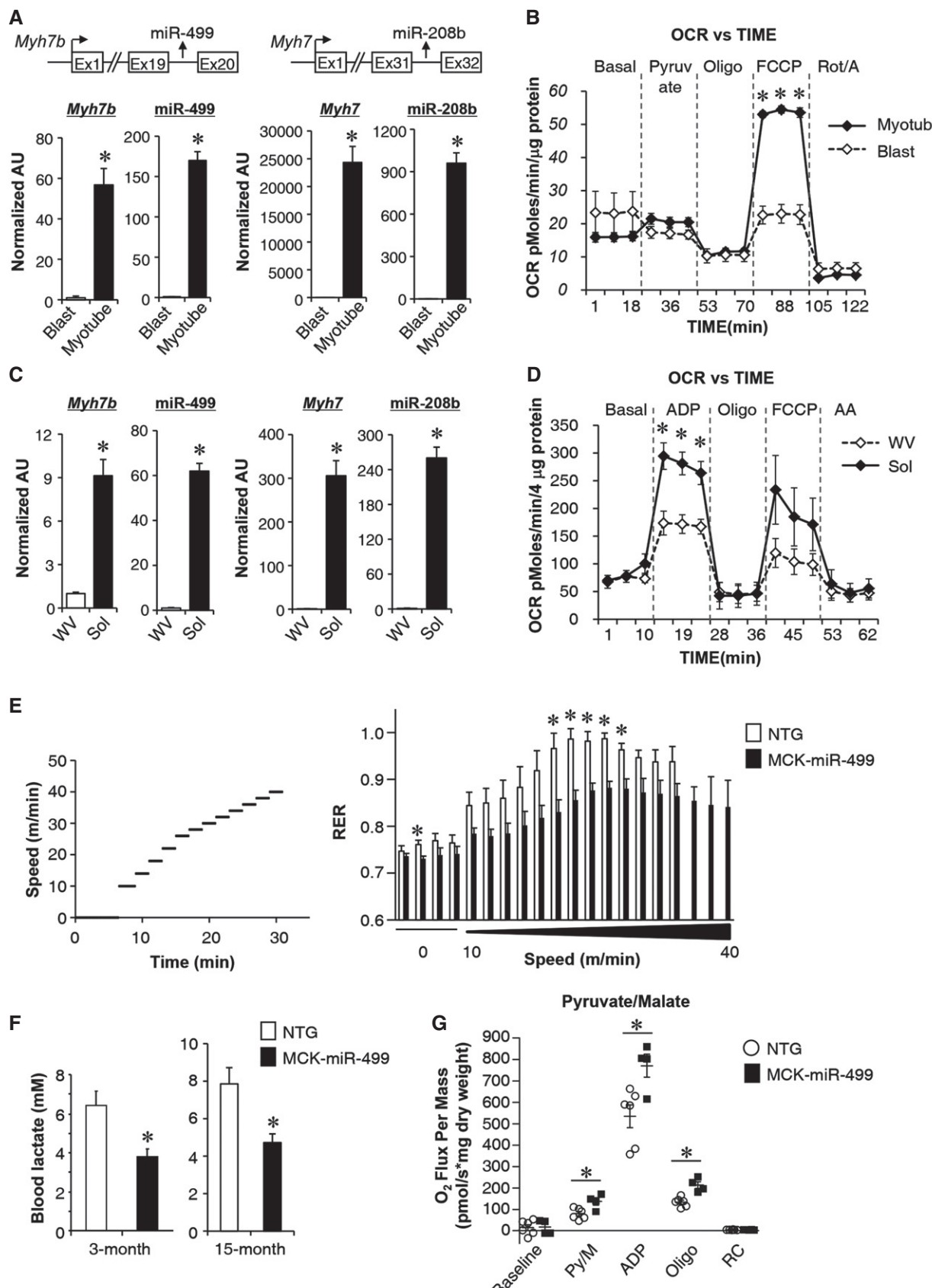

Figure 1.

**Figure 1.  miR-499 is dynamically regulated during muscle fiber type transition and affects mitochondrial function.**

A    RT–qPCR analysis of *Myh7b/miR-499* and *Myh7/miR-208b* levels during differentiation of myoblasts into mature myotubes (*n* = 3 independent experiments).
*P* = 0.0005 (*Myh7b*), *P* < 0.0001 (*miR-499*), *P* = 0.0002 (*Myh7*), *P* < 0.0001 (*miR-208b*).

B    Oxygen consumption rates (OCR) in primary myoblasts and differentiated myotubes. Basal OCR was first measured, followed by administration of 10 mM sodium
pyruvate, and 2 μM oligomycin (to inhibit ATP synthase), uncoupler FCCP (2 μM), or rotenone/antimycin (Rot/A; 1 μM) as indicted. *n* = 3 separate experiments done
with 10 biological replicates. *P* < 0.0001 (FCCP).

C    Mean expression levels (RT–qPCR) in white vastus (WV) and soleus (Sol) muscle from 8-week-old male wild-type mice (*n* = 5 mice per group). *P* < 0.0001 (*Myh7b*),
*P* < 0.0001 (*miR-499*), *P* < 0.0001 (*Myh7*), *P* < 0.0001 (*miR-208b*).

D    Mitochondrial respiration rates were determined from mitochondria isolated from indicated muscle using pyruvate as substrate. ADP-dependent respiration,
oligomycin-induced (oligo), uncoupler FCCP, and antimycin A (AA) are shown. *n* = 3 separate experiments done with 7–8 biological replicates. *P* < 0.05 (ADP).

E    (Left) Schematic depicts the increments of speed over time. (Right) Respiratory exchange ratio (RER) during a graded exercise regimen as described in Materials and
Methods (*n* = 5 mice per group). Notably, MCK-miR-499 mice were able to exercise at a higher speed before exhaustion. *P* < 0.05.

F    Bars represent mean blood lactate levels for 3- and 15-month-old male MCK-miR-499 and NTG mice following a 25-min run on a motorized treadmill. For 3-month
blood lactate, NTG, *n* = 9; MCK-miR-499, *n* = 10. For 15-month blood lactate, NTG, *n* = 7; MCK-miR-499, *n* = 5. *P* = 0.039 (3 months), *P* = 0.018 (15 months).

G    Mitochondrial respiration rates were determined from the plantaris muscle of the indicated genotypes using pyruvate/malate as substrate. Pyruvate/malate (Py/M)-
stimulated, ADP-dependent respiration, oligomycin-induced (oligo), and the respiratory control ratio (RC) are shown. NTG, *n* = 6; MCK-miR-499, *n* = 4. *P* = 0.015
(Py/M), *P* = 0.017 (ADP), *P* = 0.001 (Oligo).

Data information: All values represent the mean ± SEM. *P*-value was determined using two-tailed unpaired Student's *t*-test.

## miR-499 regulates muscle mitochondrial oxidative capacity through PGC-1α

We next sought to explore the targets and pathways downstream of miR-499 that are involved in regulation of muscle mitochondrial function. Using Ingenuity Pathways Analysis (http://www.ingenuity. com), we found that many of the genes upregulated in MCK-miR-499 muscle are known targets of PGC-1α (Fig EV1A), a master transcriptional co-regulator of mitochondrial function (Finck & Kelly, 2006; Handschin & Spiegelman, 2006). Real-time PCR confirmed that *Ppargc1a* (PGC-1α) mRNA levels were higher and *Ppargc1b* (PGC-1β) mRNA levels were lower in MCK-miR-499 muscle (Fig EV1B). *Esrrb* (ERRβ) mRNA was also modestly increased in MCK-miR-499 muscle, whereas *Esrra* (ERRα), *Esrrg* (ERRγ), *Ppara* (PPARα), and *Ppard* (PPARδ) gene expressions were not induced in MCK-miR-499 muscle (Fig EV1B). Remarkably, PGC-1α protein levels were dramatically induced (14-fold) in the GC muscle of MCK-miR-499 mice compared to NTG controls (Fig 3A). Levels of PGC-1α protein were further compared across multiple muscle types in NTG and MCK-miR-499 mice. As shown in Fig EV1C, the protein levels of PGC-1α and myoglobin were also increased in WV muscle in MCK-miR-499 mice. Interestingly, there is no increase in mitochondrial DNA levels in MCK-miR-499 muscle compared with NTG controls (Fig EV1D), suggesting that PGC-1α affects mitochondrial respiration capacity without changes in mitochondrial content in MCK-miR-499 muscle.

To determine whether the muscle mitochondrial oxidative changes induced by miR-499 were due to the marked induction of PGC-1α protein, we bred MCK-miR-499 mice with muscle-specific PGC-1α KO (PGC-1α mKO) mice to generate 499Tg/PGC-1α mKO mice, in which the PGC-1α gene is disrupted in muscle in a miR-499 transgenic background. Upon gross examination, disruption of PGC-1α markedly reduced the red coloration of MCK-miR-499 muscle (Fig EV2A and B). miR-499-mediated induction of oxidative biomarker genes (*Ldhb*, *Cox5a*, myoglobin, and cytochrome c) was significantly reduced in the absence of PGC-1α (Fig 3B and C). Furthermore, muscle-specific disruption of PGC-1α resulted in marked reduction of oxidative mitochondrial enzyme SDH activity in the GC muscle of MCK-miR-499 mice (499Tg, 56.9 ± 2.0% versus 499Tg/PGC-1α mKO, 35.4 ± 7.9%, *n* = 4 mice per group, *P* = 0.012) (Fig 3D). Interestingly, however, no change in GPDH

staining and MHC1 immunofluorescence and was observed in the GC muscle of 499Tg/PGC-1α mKO mice compared to MCK-miR-499 mice (GPDH: 499Tg, 34.3 ± 2.0% versus 499Tg/PGC-1α mKO, 41.0 ± 3.5%, *n* = 4 mice per group, *P* = 0.114) (MHC1: 499Tg, 347 ± 35 per section versus 499Tg/PGC-1α mKO, 463 ± 55 per section, *n* = 5 mice per group, *P* = 0.113) (Fig 3D). Similar observations were made in multiple muscle types of the 499Tg/PGC-1α mKO mice, in which the expression of slow-twitch muscle fiber genes was activated by miR-499 but not affected by disruption of PGC-1α in all soleus, GC, and WV muscle (Fig EV2C). Together, these results demonstrate that PGC-1α is required for the miR-499-mediated increase in mitochondrial oxidative capacity in muscle but not fiber type. It is possible that Sox6 inhibition (Fig EV2C) acts in parallel with PGC-1α signaling, thereby inducing slow myofiber formation in MCK-miR-499 mice.

We also assessed the physiological effect of PGC-1α gene disruption in MCK-miR-499 mice. As shown in Fig 3E, disruption of PGC-1α abolished the miR-499-mediated enhancement of exercise capacity and the RER-lowering effect during exercise. In addition, muscle-specific disruption of PGC-1α prevented the miR-499-mediated reduction of blood lactate levels following exercise (Fig 3F). Together, these results indicate that PGC-1α is required for the miR-499-mediated increase in oxidative metabolism in muscle.

## miR-499 directly targets Fnip1, an AMPK interacting protein that negatively regulates AMPK-PGC-1α signaling and mitochondrial function

We next sought to determine the mechanism whereby PGC-1α-dependent mitochondrial function was induced in the MCK-miR-499 muscle. miRNAs inhibit mRNA translation or promote mRNA degradation through putative basepair interactions with target mRNAs (Bartel, 2004, 2009; Chen *et al*, 2009; Mendell & Olson, 2012). Thus, bioinformatics prediction combined with 3'UTR reporter assays were used to identify putative target mRNAs for miR-499 (Fig EV3). The TargetScan and MicroCosm programs were first used to identify putative target mRNAs for miR-499, and this gene list was next cross-matched for genes that were downregulated in MCK-miR-499 muscle (Fig EV3 and Appendix Table S3). Putative targets were then further selected for additional 3'UTR reporter validation assays

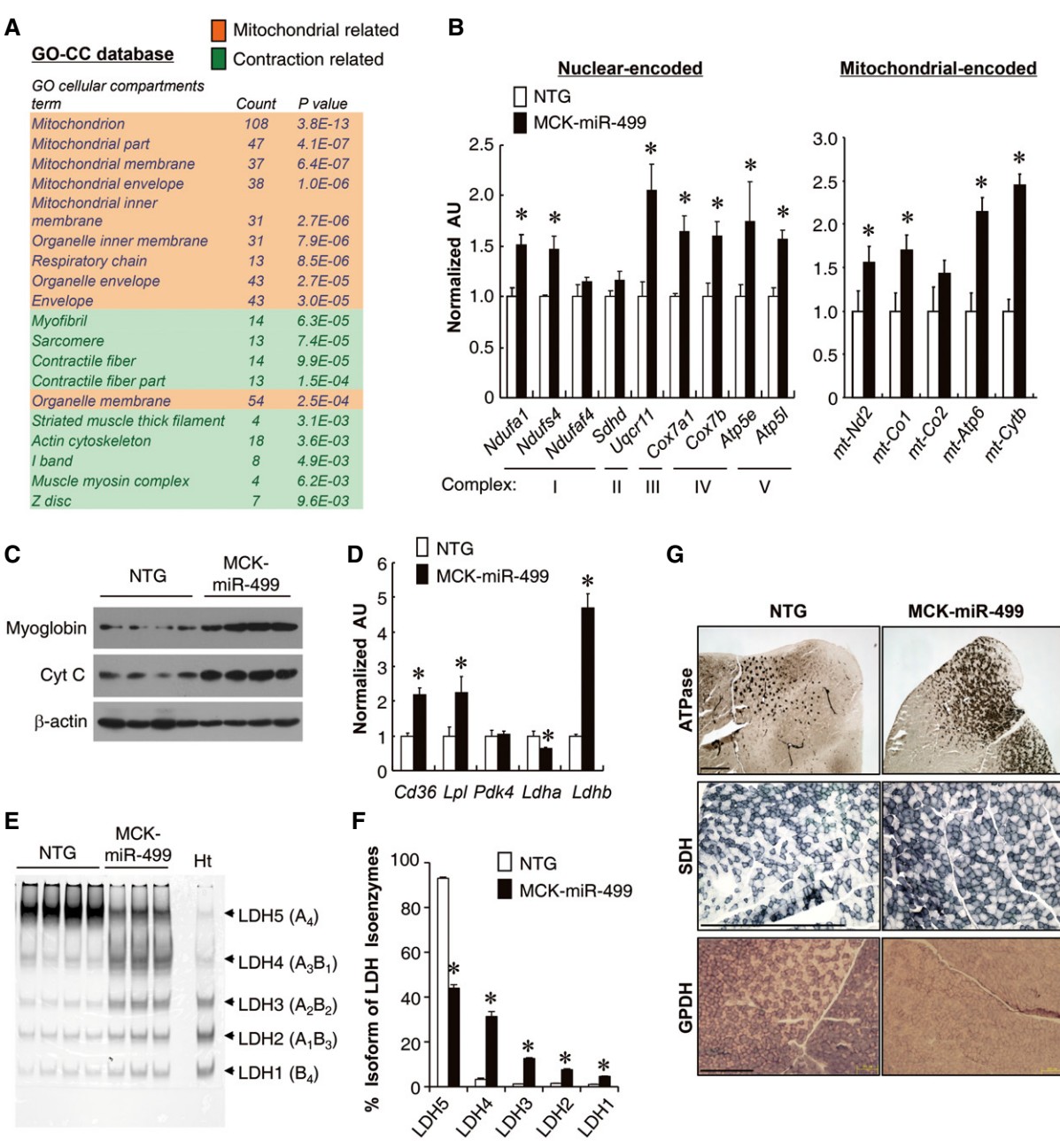

**Figure 2. MCK-miR-499 muscle is reprogrammed for increased capacity for mitochondrial oxidation.**

A  Gene ontology (GO) enrichment analysis of gene transcripts upregulated in MCK-miR-499 muscle identified a number of terms related to mitochondrial function and muscle contraction. Gene expression array data generated from the gastrocnemius muscle of 8-week-old male MCK-miR-499 mice were compared to littermate controls (NTG).

B  Expression of the nuclear-encoded and mitochondrial-encoded genes (RT-qPCR) in the gastrocnemius muscle from the indicated genotypes (n = 5 mice per group). *P < 0.05.

C  Representative Western blot analysis performed with gastrocnemius muscle total protein extracts prepared from the indicated mice using myoglobin, cytochrome c, and β-actin (control) antibodies (n = 8 mice per group).

D  Expression of genes (RT-qPCR) involved in muscle glucose and fatty acid metabolism in the gastrocnemius muscle from the indicated genotypes (n = 5 mice per group). *P < 0.05.

E  LDH isoenzymes were separated by polyacrylamide gel electrophoresis using whole cell extracts from NTG heart (Ht, control) and gastrocnemius muscle from the indicated mice. A representative gel showing 3–4 independent mice per group is shown.

F  Quantification of LDH isoenzyme activity gel electrophoresis in (E). Values represent the mean % (± SEM) total LDH activity. *P < 0.05.

G  Cross section of the gastrocnemius muscle from 3-month-old male NTG and MCK-miR-499 mice stained for myosin I ATPase activity (top), SDH (middle), and α-GPDH (bottom). Representative images are shown. Scale bar: 500 μm.

Data information: All values represent the mean ± SEM. P-value was determined using two-tailed unpaired Student's t-test.
Source data are available online for this figure.

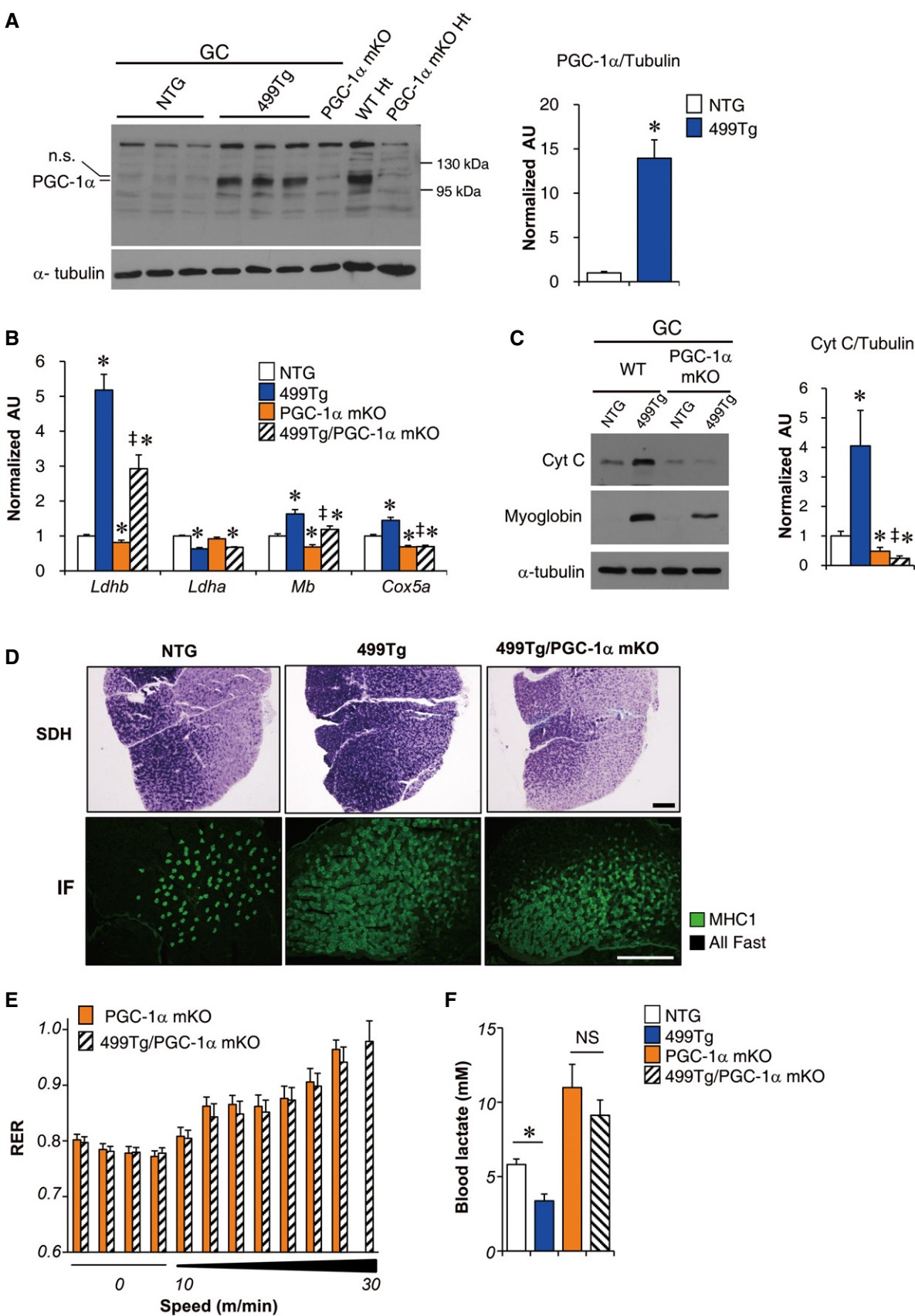

**Figure 3.**

◄

**Figure 3.  miR-499 regulates muscle mitochondrial oxidative capacity through PGC-1α.**

A   (Left) Representative Western blot analysis of PGC-1α protein expression in the gastrocnemius muscle from the indicated genotypes, heart (Ht) lysate from WT mice and PGC-1α mKO (MCK-Cre) mice as positive and negative controls, respectively. (Right) Quantification of the PGC-1α/tubulin signal ratios normalized (= 1.0) to the NTG control (n = 5 mice per group). *P = 0.00017.

B   Expression of the *Ldhb*, *Ldha*, *Mb* (myoglobin), and *Cox5a* genes (RT-qPCR) in the gastrocnemius muscle from the indicated genotypes (n = 5 mice per group). *P < 0.001 (versus NTG), ‡P < 0.01 (versus 499Tg).

C   (Left) Representative Western blot analysis performed with gastrocnemius muscle total protein extracts prepared from the indicated mice using cytochrome c, myoglobin, and α-tubulin (control) antibodies. (Right) Quantification of the Cyt c/tubulin signal ratios normalized (= 1.0) to the NTG control. NTG, n = 5; 499Tg, n = 6; PGC-1α mKO, n = 5; 499Tg/PGC-1α mKO, n = 5. *P < 0.01 (versus NTG), ‡P = 0.00018 (versus 499Tg).

D   Cross section of gastrocnemius muscle from the indicated mice stained for SDH activity (top) as well as MHC1 immunofluorescence (IF; bottom); representative images are shown. Scale bar: 500 μm.

E   Respiratory exchange ratio (RER) during a graded exercise regimen as described in Materials and Methods. Notably, no significant increase in exhaust speed was observed in 499Tg/PGC-1α mKO mice compared to PGC-1α mKO mice. PGC-1α mKO, n = 5; 499Tg/PGC-1α mKO, n = 8.

F   The bars represent the mean blood lactate levels from the indicated mice following a 25-min run on a motorized treadmill. NTG, n = 9; 499Tg, n = 7; PGC-1α mKO, n = 6; 499Tg/PGC-1α mKO, n = 9. *P = 0.0009 (NTG versus 499Tg), P = 0.316 (NS, not significant).

Data information: All values represent the mean ± SEM. P-value in (A and F) was determined using two-tailed unpaired Student's t-test; P-value in (B and C) was determined using one-way ANOVA coupled to a Fisher's least-significant difference (LSD) post hoc test.
Source data are available online for this figure.

(Fig EV3). This analysis revealed that the transcript encoding folliculin-interacting protein-1 (Fnip1) is a potential direct target of miR-499. This finding was of interest because Fnip1 was originally identified as an AMPK-interacting protein (Baba *et al*, 2006), suggesting a role in the AMPK–PGC-1α regulatory pathway. In addition, mice with genetic deletion of *Fnip1* have recently been shown to increase PGC-1α signaling and oxidative muscle fiber composition (Hasumi *et al*, 2015; Reyes *et al*, 2015), a phenotype that substantially recapitulates that of MCK-miR-499 mice. Thus, additional studies were focused on Fnip1.

As shown in Fig 4A, the miR-499 binding sequences in the *Fnip1* 3′UTR are highly conserved in mammals. miR-499 repressed an *Fnip1* 3′UTR reporter via the conserved miR-499 binding site in HEK293T cells (Fig 4B). In addition, expression of *Fnip1*, but not the related *Fnip2*, was decreased in MCK-miR-499 GC muscle compared to NTG controls (Fig 4C). Fnip1 protein levels were dramatically decreased in MCK-miR-499 GC muscle compared to NTG controls (Fig 4D). These results demonstrate that miR-499 directly targets Fnip1.

To directly determine the role of Fnip1 in miR-499-mediated induction of mitochondrial function, loss-of-function experiments were conducted in primary myotubes. siRNA-mediated knockdown of *Fnip1* resulted in a marked increase in phosphorylated AMPKα (p-AMPKα) levels (Fig 5A). In addition, levels of PGC-1α transcript were increased in myotubes treated with *Fnip1* siRNA (Fig 5B). To determine whether AMPK signaling is involved in this mechanism, the effect of AMPK inhibitor compound C was assessed together in the presence of *Fnip1* siRNA-mediated knockdown. As shown in Fig 5C, AMPK inhibition abolished the *Fnip1* siRNA-mediated induction of *Ppargc1a* mRNA. These results further establish the importance of AMPK in this mechanism, were consistent with increased AMPK activity (Fig 5D), and elevated PGC-1α levels (Fig 3A) in MCK-miR-499 muscle. Oxygen consumption rates (OCR) were measured in skeletal myotubes following knockdown of *Fnip1* expression. Fnip1 inhibition significantly stimulated OCR in the presence of the uncoupler FCCP, a sign of enhanced mitochondrial function (Fig 5E). These results demonstrate that Fnip1 is a negative regulator of mitochondrial function in myocyte.

To assess the requisite role of miR-499 in the control of mitochondrial function in the absence of overexpression and determine whether Fnip1 is involved in this mechanism, miR-499 and Fnip1 loss-of-function studies were conducted in wild-type (WT) mouse primary skeletal myotubes. Because miR-499 and miR-208b function redundantly, the impact of antisense-mediated inhibition of both miR-499 and miR-208b on *Ppargc1a* and *Fnip1* gene expression was evaluated in WT myotubes. Consistent with miR-499 suppression of Fnip1, inhibition of both miRNAs modestly reduced *Ppargc1a* mRNA levels and, conversely, induced *Fnip1* mRNA levels (Fig 5F). The effects of miR-499/miR-208b inhibition on myocyte OCR were next assessed alone and together with the presence of *Fnip1* siRNA. As shown in Fig 5G, inhibition of both miRNAs resulted in significant decrease in maximal OCR in the presence of the uncoupler FCCP, and *Fnip1* knockdown prevented the miR-499/miR-208b inhibition-mediated repression of myocyte OCR. These results further establish the relevance of miR-499-mediated regulation of mitochondrial function and demonstrate the importance of Fnip1 suppression in this mechanism. Together, our data support a model in which miR-499 directly targets Fnip1, activating AMPK-PGC-1α signaling, and thereby driving increased muscle mitochondrial capacity.

**Restoring the expression of miR-499 activates the slow-oxidative muscle fiber program and ameliorates muscular dystrophy in mdx mice**

Fast-glycolytic myofibers are more sensitive to the dystrophic pathology in Duchenne muscular dystrophy (DMD) patients as well as in the preclinical mdx mouse model of DMD (Webster *et al*, 1988; Kuznetsov *et al*, 1998; Timmons *et al*, 2005; Ljubicic *et al*, 2014). Accordingly, promoting the slow-oxidative muscle fiber program was proposed as a strategy to protect against DMD (Ljubicic *et al*, 2014). Therefore, we examined the regulation of the miR-499 circuit in mdx mice, a disease model of DMD that is known to have less slow-twitch myofibers (Jeanson-Leh *et al*, 2014; Ljubicic *et al*, 2014). As expected, the levels of miR-499 and miR-208b were significantly downregulated in mdx muscle compared to WT controls (Fig 6A). The diminished mitochondrial respiration capacity in mdx muscle was associated with decreased expression of miR-499 and miR-208b (Fig 6A and B). These results are consistent with miR-499 regulating mitochondrial function.

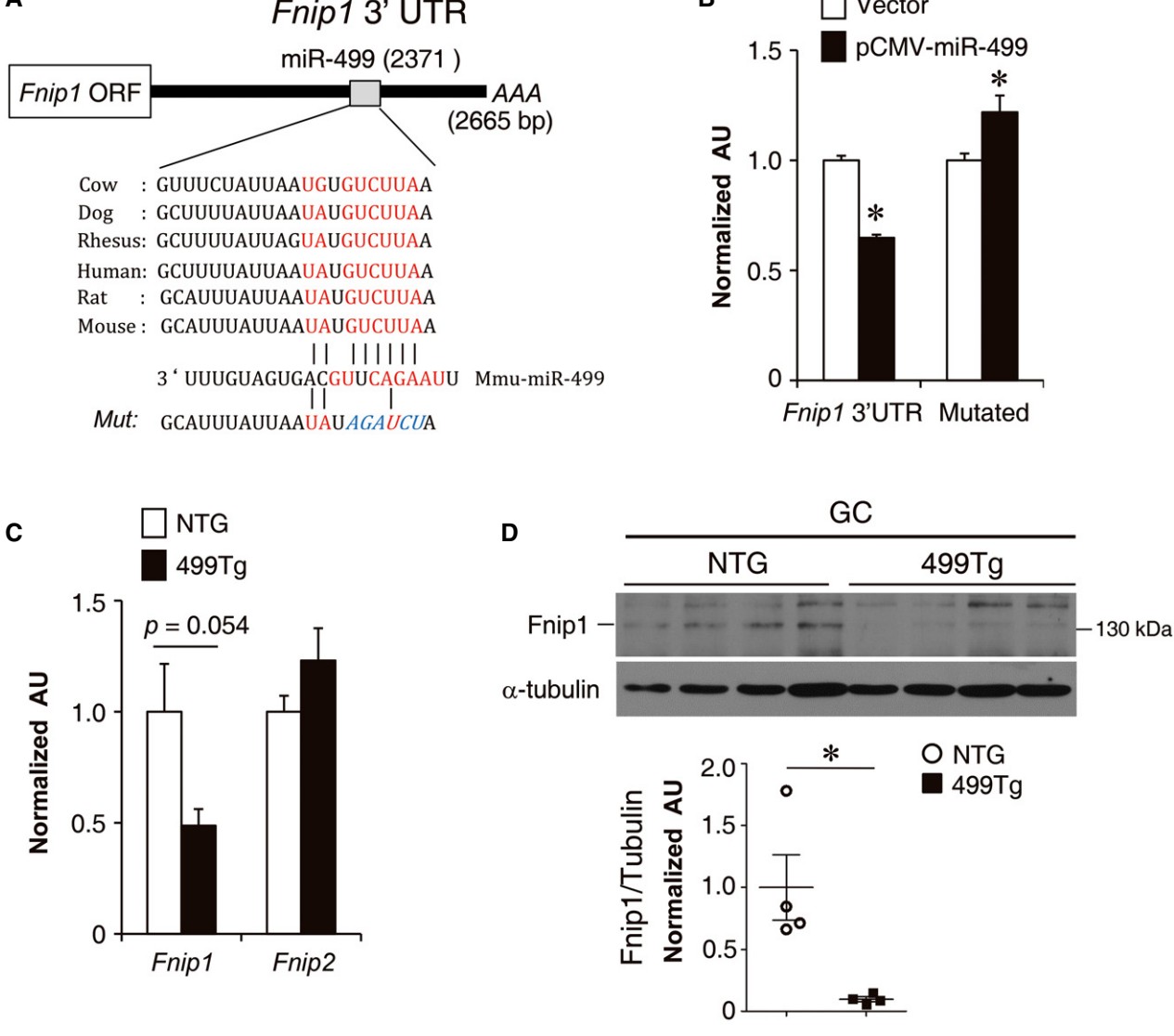

**Figure 4.  miR-499 directly targets *Fnip1*.**

A   The schematic shows the putative conserved miR-499 binding site within the 3′UTR of the *Fnip1* gene.

B   Luciferase reporters containing the wild-type *Fnip1* 3′UTR or *Fnip1* 3′UTR mutated in the predicted binding site of miR-499 were used in cotransfection studies in HEK293T cells in the presence or absence of plasmids expressing miR-499 ($n$ = 3 independent experiments). *$P$ < 0.0001 (*Fnip1* 3′UTR); *$P$ = 0.023 (Mutated).

C   RT–qPCR analysis of *Fnip1* and *Fnip2* mRNA levels in the gastrocnemius muscle of the indicated genotypes ($n$ = 5 mice per group).

D   (Top) Fnip1 protein expression in the gastrocnemius muscle from the indicated mice. (Bottom) Quantification of the Fnip1/tubulin signal ratio ($n$ = 4 mice per group). *$P$ = 0.014.

Data information: All values represent the mean ± SEM. *P*-value was determined using two-tailed unpaired Student's *t*-test.
Source data are available online for this figure.

We next determined whether transgenic restoration of the expression of miR-499 could reduce the severity of the dystrophic pathology in mdx mice. MCK-miR-499 mice were bred with the mdx mice to obtain mdx and mdx/499Tg mice. Whereas no difference in PGC-1α protein levels was observed in the GC muscle of mdx mice compared to WT controls, probably due to very low basal PGC-1α levels, activation of miR-499 in mdx muscle resulted in dramatic increase in expression of PGC-1α (Fig 6C). Fnip1 protein showed a trend toward increase in mdx muscle but significantly suppressed in mdx/499Tg muscle (Fig 6C). Interestingly, we found an increase in

phosphorylated AMPKα (p-AMPKα) levels in mdx muscle and this is reversed by miR-499 activation (Fig EV4A). The activation of p-AMPKα is consistent with cellular sensing of relative energetic deficiency within mdx dystrophic myofibers, the correction of AMPK activity in mdx/miR-499Tg muscle could also reflect a restored energy production capacity. The expression of myoglobin that was repressed in mdx mice was completely restored by miR-499 overexpression (Fig 6D). In addition, miR-499 activation also reversed oxidative biomarkers such as *Ldhb* expression in mdx muscle. The expected LDH isoenzyme activity shifts were confirmed

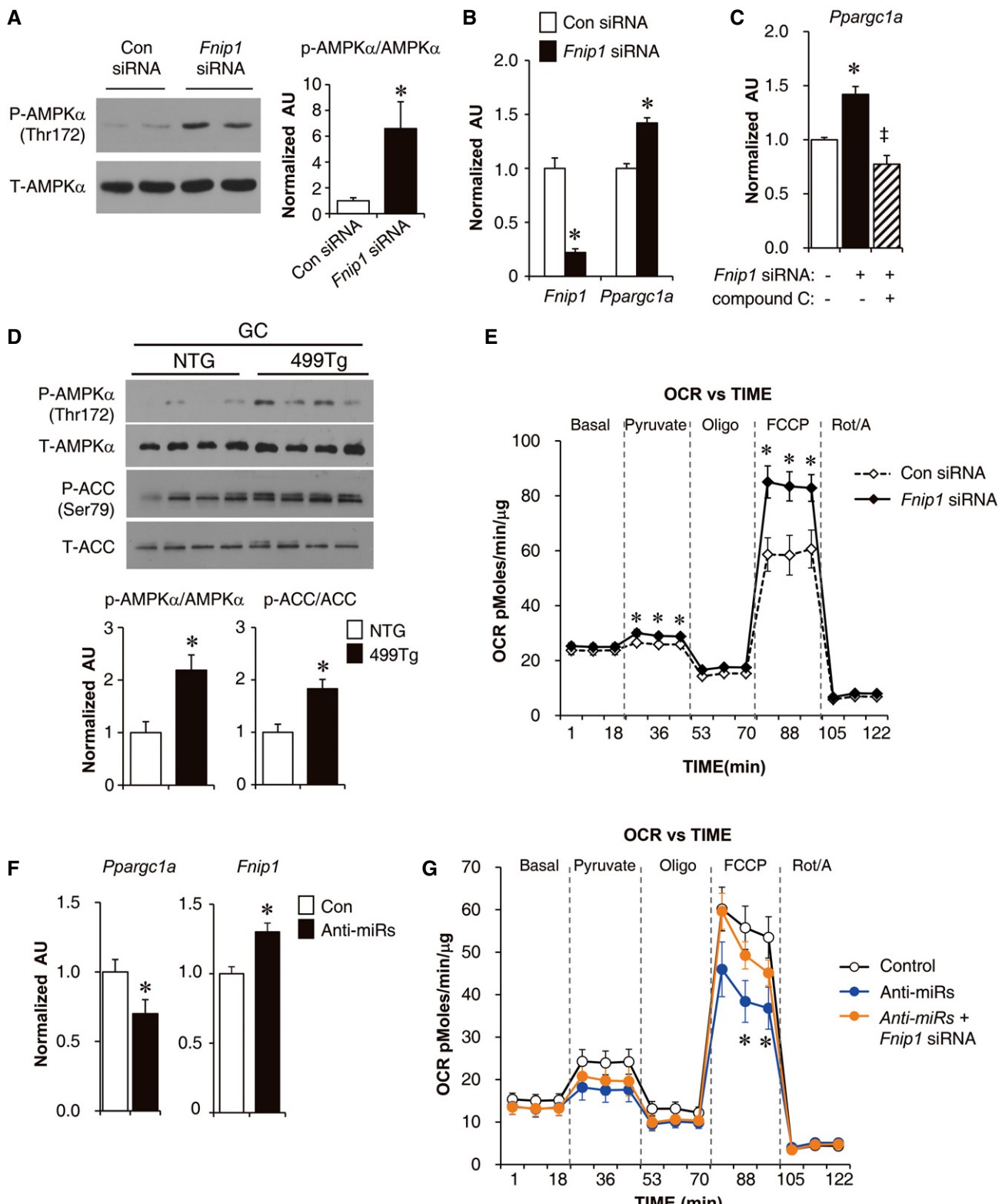

Figure 5.

by activity gel studies (Fig EV4B and C). As predicted, mitochondrial respiration capacity was significantly increased in mdx/miR-499 muscle compared to mdx mice (Fig EV4D). Interestingly,

the mitochondrial DNA levels remains unchanged in mdx/miR-499 muscle compared to mdx mice (Fig EV4E). MHC1 immunostaining demonstrated that miR-499 activation reversed the loss of type I

**Figure 5.  Inhibition of Fnip1 reactivated AMPK-PGC-1α signaling and mitochondrial function in myocytes.**

A    (Left) Representative Western blot analysis performed on extracts of myotubes subjected to *Fnip1* siRNA or control (Con) siRNA using phospho-AMPKα (Thr172) and AMPKα antibodies. (Right) Quantification of the p-AMPKα/AMPKα signal ratios ($n = 3$ independent experiments). *$P = 0.037$.

B, C  Results of RT–qPCR analysis on WT primary mouse myotubes after transfection with *Fnip1* siRNAs or scrambled Con siRNA as indicated. For (C), 48 h post-siRNA transfection, myotubes were treated for 24 h with DMSO or 10 μm compound C before harvest ($n = 3$ independent experiments). *$P < 0.0001$ (*Fnip1* in B), *$P = 0.0001$ (PGC-1α in B); *$P < 0.0001$ (versus Con siRNA in C), ‡$P < 0.0001$ (versus *Fnip1* siRNA in C).

D    (Top) Representative Western blot analysis performed on extracts of the gastrocnemius muscle isolated from NTG or MCK-miR-499 mice using phospho-AMPKα (Thr172), AMPKα, phospho-ACC (Ser79), or total ACC antibodies. (Bottom) Quantification of the p-AMPKα/AMPKα and p-ACC/ACC signal ratios normalized (= 1.0) to the NTG control ($n = 8$ mice per group). *$P < 0.01$.

E    Oxygen consumption rates (OCR) in primary mouse myotubes transfected with *Fnip1* siRNA or Con siRNA. $n = 7$ separate experiments done with 5 biological replicates. *$P < 0.05$ (Pyruvate), *$P < 0.01$ (FCCP).

F    Results of RT–qPCR analysis for WT primary mouse myotubes subjected to inhibition of both miR-499 and miR-208b (anti-miRs) ($n = 4$ independent experiments). *$P = 0.037$ (*Ppargc1a*), *$P = 0.0019$ (*Fnip1*).

G    Oxygen consumption rates (OCR) in primary mouse myotubes transfected with miR-499/miR-208b inhibitors alone and together with the presence of *Fnip1* siRNA. $n = 4$ separate experiments done with 5 biological replicates. *$P < 0.01$ (anti-miRs versus Control).

Data information: All values represent the mean ± SEM. *P*-value in (A, B, D–F) was determined using two-tailed unpaired Student's *t*-test; *P*-value in (C and G) was determined using one-way ANOVA coupled to a Fisher's least-significant difference (LSD) *post hoc* test.
Source data are available online for this figure.

muscle fibers in mdx muscle (Fig 6E). These results demonstrate that transgenic induction of miR-499 restored the slow-oxidative muscle fiber program in mdx mice.

We next characterized the dystrophic phenotype of the mdx/499Tg mice. Mdx mice had a higher variation in fiber size and a higher number of small fibers compared to WT mice. The mdx/miR-499Tg mice had a lower variation in fiber size, and had fewer small fibers compared to mdx alone (Fig EV5A). Muscle fibers from mdx mice contain higher numbers of centralized nuclei. Transgenic induction of miR-499 in mdx mice resulted in a significant reduction in the percentage of centrally located nuclei (mdx, $55.0 ± 3.4\%$; mdx/499Tg, $42.1 ± 3.3\%$; $n = 5$ mice per group, $P = 0.031$) (Figs 6E and EV5B). Evans blue dye infiltration test also showed a decrease in damaged myofibers in tibialis anterior (TA) muscle of mdx/499Tg compared with mdx mice (Fig EV5C). Additionally, we also measured the levels of serum creatine kinase (CK), another hallmark of damaged muscles, in young WT, mdx, and mdx/499Tg mice. As expected, we observed a marked increase in CK levels in mdx mice compared to WT controls. Remarkably, transgenic induction of

miR-499 in mdx mice resulted in a dramatic reduction of CK levels (mdx, $18,754 ± 4,330$ U/l; mdx/499Tg, $5,832 ± 474$ U/l; mdx, $n = 10$; mdx/499Tg, $n = 14$; $P = 0.0003$) (Fig 6F).

To assess muscle functionality and performance, we challenged mdx and mdx/499Tg mice with treadmill exercise. The mdx mice ran on average 475 m, and the mdx/499Tg mice ran significantly farther, 1,307 m, a distance indistinguishable from that run by WT controls (Fig 6G). We further conducted real-time RER measurements to evaluate the muscle fuel utilization of 499Tg/mdx mice during exercise. In mdx mice, a rapid increase in RER values occurred at a very low speed during the exercise challenge, which is consistent with a marked decrease in exercise capacity (Fig 6H). Conversely, 499Tg/mdx mice were able to exercise at much higher speeds before an increase in RER values was observed (Fig 6H), suggesting that 499Tg/mdx mice restored the capacity to utilize fatty acid oxidation during exercise. Together, these results strongly suggest that re-activation of miR-499 in mdx muscle significantly improves the pathophysiology of the muscular dystrophic phenotype.

**Figure 6.  Restoring the expression of miR-499 activates the slow-oxidative muscle fiber program and ameliorates muscular dystrophy in mdx mice.**

A    Mean expression levels (RT–qPCR) in gastrocnemius muscle of 8-week-old male WT and mdx mice ($n = 5$ mice per group). *$P = 0.009$ (*Myh7b*), *$P = 0.0007$ (*miR-499*), *$P = 0.0143$ (*Myh7*), *$P < 0.0011$ (*miR-208b*).

B    Mitochondrial respiration rates were determined from mitochondria isolated from the hindlimb muscle of the indicated mice using pyruvate as substrate. $n = 3$ separate experiments done with 7–8 biological replicates. *$P < 0.01$ (ADP).

C    (Left) Representative Western blot analysis of PGC-1α (Top) and Fnip1 (Bottom) protein expression in the gastrocnemius muscle from the indicated genotypes with α-tubulin as the loading control. (Right) Quantification of the PGC-1α/tubulin and Fnip1/tubulin signal ratios normalized (= 1.0) to the WT control. WT, $n = 8$; mdx, $n = 5$; mdx/499Tg, $n = 7$. *$P < 0.0001$ (versus WT), ‡$P < 0.0001$ (versus mdx).

D    Representative Western blot analysis of myoglobin protein expression in the gastrocnemius muscle from the indicated genotypes with α-tubulin as the loading control ($n = 5$ mice per group).

E    (Top) Representative MHC1 immunofluorescence (IF) in the soleus of the indicated genotypes. Scale bar: 500 μm. (Bottom) Representative H&E staining of the gastrocnemius muscle of the indicated genotypes. Scale bar: 100 μm.

F    Five-week-old male WT, mdx, and mdx/MCK-miR-499 mice were euthanized. and serum creatine kinase activity was determined. WT, $n = 10$; mdx, $n = 10$; mdx/499Tg, $n = 14$. *$P < 0.0001$ (versus WT), ‡$P = 0.0003$ (versus mdx).

G    The bars represent the mean running time and distance (± SEM) for 8-week-old male mice on a motorized treadmill. WT, $n = 10$; mdx, $n = 13$; mdx/499Tg, $n = 8$. *$P = 0.01406$ (Running time, versus WT), ‡$P = 0.005756$ (Running time, versus mdx). *$P = 0.004558$ (Running distance, versus WT), ‡$P = 0.001668$ (Running distance, versus mdx).

H    Respiratory exchange ratio (RER) during a graded exercise regimen on a motorized treadmill. mdx, $n = 8$; mdx/499Tg, $n = 6$. ‡$P < 0.05$.

Data information: All values represent the mean ± SEM. *P*-value in (A, B and H) was determined using two-tailed unpaired Student's *t*-test; *P*-value in (C, F and G) was determined using one-way ANOVA coupled to a Fisher's least-significant difference (LSD) *post hoc* test.
Source data are available online for this figure.

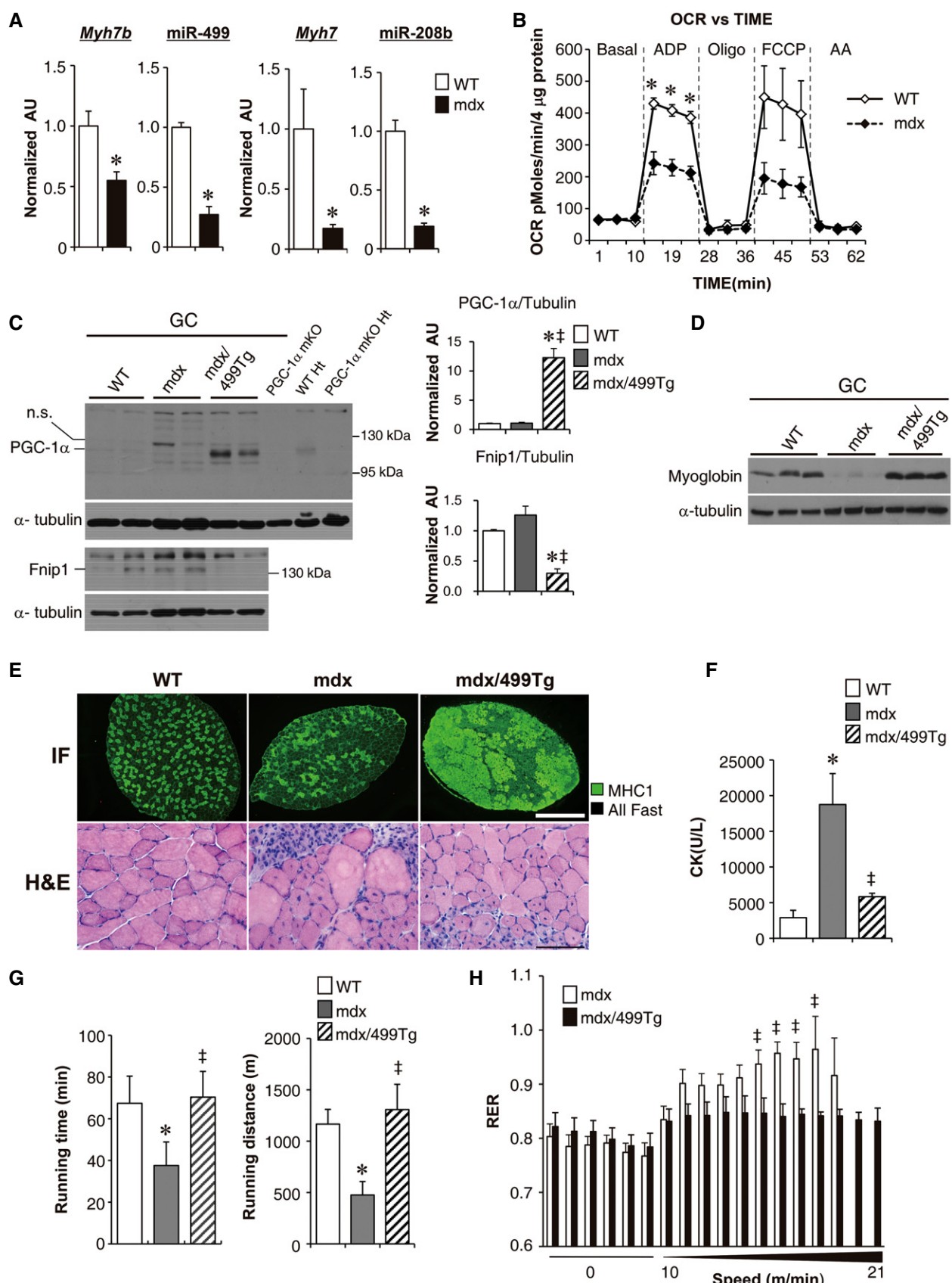

Figure 6.

## Discussion

Delineation of the mechanisms involved in the coordinate regulation of muscle metabolic and structural programs during fiber type transition has implications for new therapeutic approaches for many human diseases, including metabolic disorders and muscular dystrophy. Previous studies have established that nuclear receptors such as PPARs and ERRs, along with their co-regulators PGC-1s and AMPK are key regulators of myocytes energy metabolism (Lin et al, 2002; Wang et al, 2004; Arany et al, 2005; Schuler et al, 2006; Narkar et al, 2008, 2011; Rangwala et al, 2010; Zechner et al, 2010; Gan et al, 2011, 2013). Herein, we discover a novel mechanism for muscle contractile property tightly coupled to its metabolic capacity during fiber type transition. Specifically, the myosin *Myh7b* gene encodes miR-499, which directly inhibits Fnip1, leading to activation of AMPK-PGC-1α signaling and thereby triggering a muscle mitochondrial oxidative metabolism program. We therefore propose a model for the adaptive mitochondrial function during muscle fiber type transition via the miR-499/Fnip1/AMPK circuit (Fig 7). This mechanism likely represents a general paradigm for efficiently couple cellular energy consumption with ATP production under an array of diverse physiological and pathophysiological circumstances.

Nuclear receptor ERRγ was recently shown to drive a slow-oxidative muscle fiber program in skeletal muscle, including increased fatty acid oxidation, mitochondrial oxidative phosphorylation, and slow-twitch muscle fiber composition (Rangwala et al, 2010; Narkar et al, 2011; Gan et al, 2013). We have recently found that ERRγ directly activates the promoters driving the expression of miR-499 and miR-208b in muscle (Gan et al, 2013). In the present study, our data suggest that miR-499 is a regulator of mitochondrial function that acts downstream of the ERRγ signaling. Exercise training induces the expression of ERRγ as well as miR-499 in muscle (Gan et al, 2013). It is likely that the miR-499/Fnip1/AMPK circuit serves as a major nexus between slow myofiber stimuli and the adaptive mitochondrial function, such as exercise training in which the transcription changes in contractile myosin genes parallels the changes in muscle metabolic properties (Kirschbaum et al, 1990; Schiaffino & Reggiani, 2011). Conversely, chronic immobilization and disuse is predicted to have the opposite effect.

We found that miR-499 regulates muscle metabolism by directly targeting Fnip1. The observed role of Fnip1 is of interest given its role in regulating muscle metabolism (Baba et al, 2006; Hasumi et al, 2015; Reyes et al, 2015). Notably, knockout of the gene encoding folliculin, an Fnip1 interacting protein, has also been shown to induce slow-oxidative muscle transformation (Hasumi et al, 2012), which further emphasizes the importance of Fnip1-related signaling in muscle metabolism. Our data strongly suggest that the major effect of miR-499 on muscle energy metabolism might be mediated through inhibition of Fnip1 protein. First, we found that Fnip1 mRNA and protein expression in muscle were markedly suppressed in MCK-miR-499 mice. Second, *Fnip1* null mice recapitulated many of the

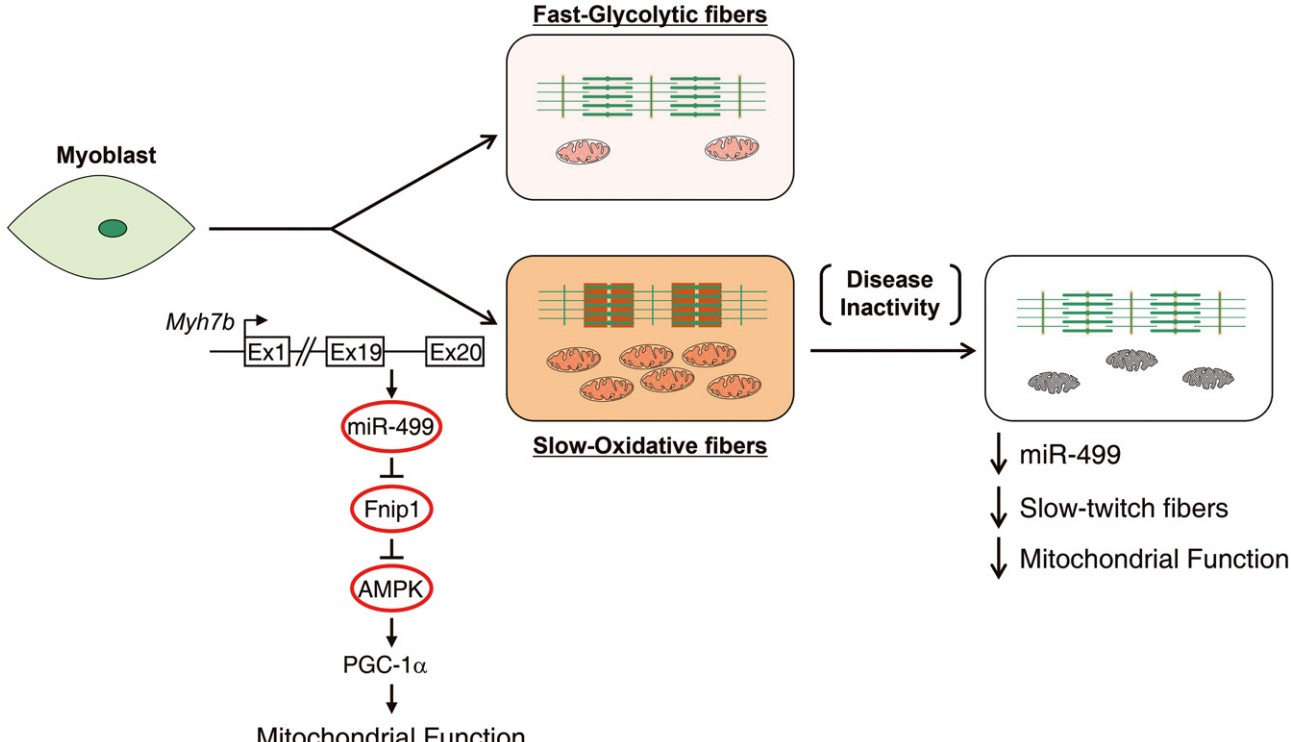

**Figure 7.  A model for the adaptive mitochondrial function during muscle fiber type transition.**
The schematic depicts a proposed model for the miR-499/Fnip1/AMPK circuit that orchestrates mitochondrial function to match muscle contractile machinery during fiber type transition.

metabolic phenotypes observed in the MCK-miR-499 mice, including enhanced red coloration, increased capacity for SDH activity and mitochondrial respiration. Third, key regulators of aerobic metabolism such as AMPK/PGC-1α and oxidative biomarkers induced in MCK-miR-499 muscle were similarly activated in *Fnip1* knockout muscle. Fourth, direct manipulation of Fnip1 in primary myotubes mimicked the effects of miR-499 on AMPK/PGC-1α levels and mitochondrial respiration in muscle. However, unlike Fnip1 (Reyes *et al*, 2015), an interesting finding of our study is that miR-499 drives an increase in slow-twitch muscle fiber proportion in a PGC1α-independent manner. It is possible that Sox6, another direct target of miR-499, acts in parallel with Fnip1/PGC-1α signaling to induce slow myofiber formation in MCK-miR-499 muscle. Consistent with this notion, a recent study has concluded that Sox6 acts independently of PGC-1α on muscle fiber type switching in mice (Quiat *et al*, 2011). Interestingly, we also found evidence that miR-499 activates glucose oxidation in muscle. The *Ldhb/Ldha* gene expression ratio is increased in MCK-miR-499 muscle. These results were of interest because the *Ldhb* isoenzyme favors the reaction that converts lactate to pyruvate which, in turn, provides substrate for the mitochondrial TCA cycle, whereas *Ldha* favors the reverse reaction to produce lactate from pyruvate. Therefore, miR-499 drives an *LDH* isoenzyme shift that diverts pyruvate into the mitochondrion to fully catabolize glucose for maximal ATP production, contributing to enhanced exercise capacity and mitochondrial activity, and decreased post-exercise blood lactate in MCK-miR-499 mice.

Fast-glycolytic myofibers are more sensitive to the dystrophic pathology in Duchenne muscular dystrophy (DMD) patients as well as in mdx mice (Webster *et al*, 1988; Kuznetsov *et al*, 1998; Timmons *et al*, 2005; Ljubicic *et al*, 2014). In this study, we found that downregulating the miR-499 circuit in the mdx muscle diminished the mitochondrial function in parallel. Based on our data, it is tempting to speculate that downregulated miR-499 signaling contributes to a vicious cycle, further promoting the progression of muscular dystrophy. Direct manipulation of miR-499 is able to restore the slow-oxidative muscle fiber program and prevent the hallmarks of DMD, including serum CK and exercise capacity. There was a strong correlation between Fnip1/PGC-1α signaling and miR-499 levels in the mdx/499Tg model. The activation of PGC-1α specifically in skeletal muscle is known to improve muscular dystrophy (Handschin *et al*, 2007; Selsby *et al*, 2012; Chan *et al*, 2014). Moreover, *Fnip1* knockout mice in an mdx genetic background have recently been shown to be protected against muscular dystrophy (Reyes *et al*, 2015). Based on our data, it is likely that the inhibition of Fnip1 leading to activation of PGC-1α signaling could contribute to the prevention of muscular dystrophy in 499Tg/mdx mice. Whereas a shift in fiber type is also a key component of the "rescue" response, the coupling of fiber type to appropriate mitochondrial function is likely necessary. Recent work has linked defects in mitophagy to mdx muscular dystrophy (Pauly *et al*, 2012), and it is also possible that miR-499 enhance muscle function by promoting the clearance of defective mitochondria.

In summary, we uncovered a mechanism for coupling of adaptive mitochondrial function to contractile fiber type via the miR-499/Fnip1/AMPK circuit, and this regulatory circuit shows therapeutic potential for maintaining muscle energetics and function in a variety of chronic disease states such as muscular dystrophy.

# Materials and Methods

## Animal studies

All animal studies were conducted in strict accordance with the institutional guidelines for the humane treatment of animals and were approved by the IACUC committees at the Model Animal Research Center (MARC) of Nanjing University. MCK-miR-499 mice (van Rooij *et al*, 2009) originally generated at UT Southwestern were back-crossed to the C57BL/6J background for more than six generations. To generate mice with a muscle-specific disruption of the PGC-1α allele in MCK-miR-499 mice (herein named 499Tg/PGC-1α mKO), PGC-1α f/f mice (Jackson Laboratory, stock no. 009666) were crossed with mice expressing Cre under control of the muscle creatine kinase (MCK) promoter (Jackson Laboratory, stock no. 006475) to achieve muscle-specific deletion of PGC-1α, and these mice were then crossed with MCK-miR-499 mice to obtain 499Tg/PGC-1α mKO mice. The mdx mice were purchased from the Jackson Laboratory (stock no. 000476), and male MCK-miR-499 mice were bred with homozygous female mdx mice to generate 499Tg/mdx mice. Male offspring were genotyped and categorized in mdx and 499Tg/mdx mice, and mice at the age of 5–9 weeks were used. Littermate controls were used in all cases. Mice were randomly assigned to various analyses. Investigators involved in the histological analysis were blinded. Investigators performing animal handling, sampling, and raw data collection were not blinded.

## Mitochondrial respiration studies

Mitochondrial respiration rates were measured in saponin-permeabilized plantaris or extensor digital longus muscle fibers with pyruvate/malate as substrates as described previously (Saks *et al*, 1998; Coen *et al*, 2013; Lai *et al*, 2014). In brief, the muscle fibers were separated and transferred to BIOPS buffer (7.23 mM K₂EGTA, 2.77 mM CaK₂EGTA, 20 mM imidazole, 20 mM taurine, 50 mM potassium 2-[N-morpholino]-ethanesulfonic acid, 0.5 mM dithiothreitol, 6.56 mM MgCl₂, 5.7 mM ATP, and 14.3 mM phosphocreatine [PCr], pH 7.1). The muscle fiber bundles were then permeabilized with 50 μg/ml saponin in BIOPS solution. Measurement of oxygen consumption in permeabilized muscle fibers was performed in buffer Z (105 mM potassium 2-[N-morpholino]-ethanesulfonic acid, 30 mM KCl, 10 mM $KH_2PO_4$, 5 mM $MgCl_2$, 5 mg/ml BSA, 1 mM EGTA, pH7.4) at 37°C and in the oxygen concentration range 220–150 nmol $O_2$/ml in the respiration chambers of an Oxygraph 2K (Oroboros Inc., Innsbruck, Austria). Following measurement of basal, pyruvate (10 mM)/malate (5 mM) respiration, maximal (ADP-stimulated) respiration was determined by exposing the mitochondria to 4 mM ADP. Uncoupled respiration was evaluated following addition of oligomycin (1 μg/ml). Respiration rates were determined and normalized to fiber bundle dry weight or wet weight using Datlab 5 software (Oroboros Inc., Innsbruck, Austria), and the data were expressed as "pmol $O_2$/s/mg dry weight" or "pmol $O_2$/s/mg wet weight".

Skeletal muscle mitochondria were isolated from hindlimbs of WT and mdx mice as previously described (Zechner *et al*, 2010). Total protein was quantified by a BCA assay (Thermo Scientific),

 

and respiration rates of the isolated mitochondria containing 4 μg protein/well were measured using a XF24 analyzer (Seahorse Bioscience Inc.) per the manufacturer's protocol. Respiration was determined using pyruvate as substrate. Following measurement of the basal respiration, maximal ADP-stimulated respiration was determined by exposing mitochondria to 4 mM ADP. Uncoupled respiration was evaluated following the addition of oligomycin (1 μM) to inhibit ATP synthase; this was followed by addition of the uncoupler FCCP (2 μM), and then the addition of antimycin A (0.2 μM). Respiration rates were expressed as "OCR pmoles/min 4 μg protein".

### Exercise studies

Mice were acclimated (run for 9 min at 10 m/min followed by 1 min at 20 m/min) to the treadmill for 2 consecutive days prior to the experimental protocol. Endurance exercise was determined as described previously (Gan *et al*, 2013). In brief, fed mice were run for 10 min at 10 m/min followed by a constant speed of 20 m/min until exhaustion. Tail blood was taken after exercise and measured for lactate (Lactate Scout, Senelab, Germany) according to manufacturer's instruction.

Respiratory exchange ratios (RER) during exercise were determined as described previously (Gan *et al*, 2011). Briefly, mice were placed in an enclosed treadmill attached to the Comprehensive Laboratory Animal Monitoring System (CLAMS) (Columbus Instruments) for 15 min at a 0° incline and 0 m/min. The mice were then challenged with two-min intervals of increasing speed at a 0° incline. The increasing speeds used in the protocol were 10, 14, 18, 22, 26, 28, 30, 32, 34, 36, 38, 40, 42, 44, and 46 m/min. Measurements were collected before the exercise challenge, throughout the challenge, and following failure. For the mdx and mdx/miR-499Tg mice running capacity test, the increasing speeds used in the protocol were 10, 14, 15, 16, 17, 18, 19, 20, 21, 22, 23, 24, and 25 m/min.

### Histological analyses

Muscle tissue was frozen in isopentane that had been cooled in liquid nitrogen. MHC ATPase and immunofluorescence (IF) stains were conducted as previously described (Zechner *et al*, 2010; Gan *et al*, 2013). MHC ATPase stains were performed at pH 4.31. Under these acidic conditions, MHC2 isoforms are inactivated whereas MHC1 is still functional, resulting in the addition of black dye to MHC1-positive muscle fibers. For IF stains, the muscle fibers were stained with antibodies directed against MHC1 (BA-D5) (MHC1, green; All fast MHC2, black [unstained]). H&E, SDH, and α-GPDH staining were performed as previously described (Zechner *et al*, 2010). Quantification of histological staining was performed in a blinded manner with Image-Pro Plus software.

### Serum creatine kinase assay

Mouse blood was collected, and the serum was isolated using heparin-coated collection tubes. Serum creatine kinase activity was then determined in a blinded fashion with a Beckman Coulter

AU5400 using the Creatine Kinase Assay Kit (Beckman Coulter) according to the manufacturer's protocol.

### Gene expression array studies

Total RNA isolated from the gastrocnemius muscle of 8-week-old male MCK-miR-499 and NTG littermate mice was used for gene expression array studies performed as previously described using an Affymetrix GeneChip Mouse Gene 1.0 ST array (Lai *et al*, 2014). The Affymetrix probe level data were processed using Robust Multi-Array Analysis (RMA algorithm using the Expression Console software Version 1.1 default settings) to obtain normalized summary scores of expression for each probe set on each array. Partek analysis using the ANOVA was used to identify genes differentially expressed in two conditions. Three independent samples per group were analyzed. The criteria for a regulated gene were a fold change greater than 1.3 (either direction) and a significant *P*-value (< 0.05) versus NTG. For pathway analysis, the filtered data sets (a total of 1,002 upregulated genes) were uploaded into DAVID Bioinformatics Resources 6.7 to review the bio pathways using the Gene Ontology database. The functional annotation analysis was used to interpret data, and the pathways were reviewed using the cellular compartments defined by Gene Ontology, which is ranked by *P*-value of a regulated term in Fig 2A. The gene expression arrays have been deposited in the NCBI Gene Expression Omnibus and are accessible through GEO Series accession number GSE72581.

### RNA analyses

Quantitative RT–PCR was performed as described previously, with modifications (Gan *et al*, 2011, 2013). Briefly, total RNA was extracted from mouse muscle or primary myotubes using RNAiso Plus (Takara Bio). Isolated total RNA integrity was electrophoretically verified by ethidium bromide staining. About 1 μg total RNA samples were then reverse transcribed with the PrimeScript RT Reagent Kit with gDNA Eraser (Takara Bio) using random hexamer primers according to the manufacturer's instructions. Real-time quantitative qPCR was performed using the ABI Prism Step-One system with SYBR® Premix Ex Taq™ (Takara Bio). A four-step experimental run protocol was used: (i) denaturation program (30 s at 95°C); (ii) amplification and quantification program repeated 40 times (5 s at 95°C; 34 s at 60°C); (iii) melting curve program (60–95°C with a heating rate of 0.3°C per 15 s and a continuous fluorescence measurement); (iv) and cooling program down to 40°C. Specific oligonucleotide primers for target gene sequences are listed in Appendix Table S4. Arbitrary units of target mRNA were corrected to the expression of *36b4*.

### Mitochondrial DNA analyses

Genomic/mitochondrial DNA was isolated using the RNAiso Plus (Takara Bio), followed by back extraction with 4 M guanidine thiocyanate, 50 mM sodium citrate, and 1 M Tris and an alcohol precipitation. Mitochondrial DNA content was determined by SYBR Green analysis (Takara Bio). The levels of NADH dehydrogenase subunit 1 (mitochondrial DNA) were normalized to the levels of lipoprotein

 

lipase (genomic DNA). The primer sequences are noted in Appendix Table S4.

## TaqMan miRNA

TaqMan methods were used to assess miRNA expression as previously described (Gan *et al*, 2013). The TaqMan MicroRNA Assays were specific for each miRNA (Assay ID: hsa-miR-208b, 2290; mmu-miR-499, 1352) or for the internal controls (Assay ID: snoRNA202, 1232) (Applied Biosystems). Thermal cycling was performed on an ABI Prism Step-One system. The relative miRNA level was corrected to snoRNA202.

## Antibodies and immunoblotting studies

Antibodies directed against MHC1 (BA-D5) (6 μg/ml) were purchased from the Developmental Studies Hybridoma Bank; antibodies directed against p-AMPKα (Thr172) (2535) (1:1,000), AMPKα (5831) (1:1,000), p-ACC (11818) (1:1,000), and ACC (3676) (1:1,000) were all from Cell Signaling; anti-Fnip1 antibody (ab61395) (1:1,000) was purchased from Abcam, anti-cytochrome c (bs1089) (1:1000) and anti-α-tubulin (bs1699) (1:5,000) antibody were from Bioworld, anti-myoglobin (sc-25607) (1:1,000) and anti-β-actin (sc-47778) (1:5,000) were from Santa Cruz, and anti-PGC-1α (1:1,000) was developed in the laboratory of Daniel Kelly as previously described (Leone *et al*, 2005). Western blotting studies were performed as previously described (Gan *et al*, 2011, 2013).

## LDH isoenzyme analysis

LDH isoenzyme patterns were determined as previously described (Gan *et al*, 2011). Protein extracted from mouse hearts served as the positive control.

## Cell culture and RNAi experiments

Primary muscle cells were isolated from skeletal muscles as previously described (Gan *et al*, 2011, 2013). For differentiation, cells were washed with PBS and re-fed with 2% horse-serum/DMEM differentiation medium and re-fed daily. siRNAs (Genepharma) targeting mouse *Fnip1* (siRNA pool, #1: 5′- GGAGAUAGUUCUUCC UCUUTT; #2 5′- GCAGUUCACAGCAACCCAATT; #3 5′- GGUGGCUA CUGCUCAUCUUTT) and microRNA inhibitors (Genepharma, anti-miR-499: AAACAUCACUGCAAGUCUUAA; anti-miR-208b: ACAAA CCUUUUGUUCGUCUUAU) were transfected into primary myoblasts at a final concentration of 50 nM using Lipofectamine 2000 transfection reagent (Invitrogen) according to the manufacturer's instructions. Cells were then differentiated for 3 days prior to harvest. For AMPK inhibition study, 48 h post-siRNA transfection, myotubes were treated for 24 h with DMSO or 10 μm compound C (Selleckchem).

## Oxygen consumption measurements

Cellular oxygen consumption rates (OCR) were measured using the XF24 analyzer (Seahorse Bioscience Inc.) per the manufacturer's protocol. The basal OCR was first measured in XF Assay

**The paper explained**

**Problem**

Increasing evidence suggests that in a variety of common disease states, and with aging, a vicious cycle develops whereby reduced physical activity leads to "de-training" of skeletal muscle which further constrains locomotor activity. The two determinants of muscle function: contractile fiber type and mitochondrial function are coordinately regulated upon adaption of muscle to physiological stimuli and systemic disease. However, the mechanisms for precise coupling of mitochondrial function and muscle contractile machinery during fiber type transition are unclear.

**Results**

We have unveiled a regulatory circuit that orchestrates muscle contractile machinery and energy production during adaption of skeletal muscle to physiological and pathophysiological stimuli. We show that miR-499, a miRNA embedded in muscle contractile gene, drives an adaptive mitochondrial function to match shifts in muscle fiber type composition. Moreover, we demonstrate that this miR-499 regulatory circuit is implicated in disease stress as evidenced by its favorable impact on the muscular dystrophy phenotype.

**Impact**

The miR-499 regulatory circuit unveiled in this study likely represents a general paradigm for the efficient coupling of cellular energy consumption with ATP production; it shows therapeutic potential for maintaining muscle energetics and function in a variety of chronic disease states such as muscular dystrophy.

Media without sodium pyruvate, followed by administration of 10 mM sodium pyruvate. Uncoupled respiration was evaluated following the addition of oligomycin (2 μM) to inhibit ATP synthase, by addition of the uncoupler FCCP (2 μM), and then followed by the addition of rotenone/antimycin (1 μM). Immediately after measurement, total protein levels were measured with the Micro BCA Protein Assay Kit (Thermo Scientific) for data correction.

## Cell transfection and luciferase reporter assays

The luciferase assay was performed as previously described, with modification (Gan *et al*, 2011). The pCMV6-miR-499 vector was kindly provided by Dr. Eric Olson (UT Southwestern). To generate the 3′UTR luciferase reporters of miR-499 putative targets, the 3′UTRs containing the predicted binding site of miR-499 were amplified by PCR and inserted into the pGL3-promoter (Promega). Site-directed mutagenesis was performed using complementary oligonucleotides as follows (with mutated nucleotides shown in lowercase): 5′GTATCTTGCATTTATTAATAT**agatct**ATTTTGAATTTGAAAACATG (Fnip1 3′UTRmut). HEK293T cells were obtained from the American Type Culture Collection, and were cultured at 37°C and 5% $CO_2$ in Dulbecco's modified Eagle's medium supplemented with 10% fetal calf serum, 1,000 U/ml penicillin, and 100 μg/ml streptomycin. Transient transfections in HEK293T cells were performed using PEI Transfection Reagent (Polysciences) following the manufacturer's protocol. Briefly, 350 ng of reporter was cotransfected with 100 ng of pCMV6-miR-499 and 25 ng of CMV promoter-driven *Renilla* luciferase to control for transfection efficiency. Cells were harvested

24 h after transfection. The luciferase assay was performed using Dual-Glo (Promega) according to the manufacturer's recommendations. All transfection data are presented as the mean ± standard error of the mean (SEM) for at least three separate transfection experiments.

### Statistical analyses

All mouse and cell studies were analyzed by Student's *t*-test (two-tailed) or one-way ANOVA coupled to a Fisher's least-significant difference (LSD) *post hoc* test when more than two groups were compared. No statistical methods were used to predetermine sample sizes. Data represent the mean ± SEM, with a statistically significant difference defined as a value of $P < 0.05$.

**Expanded View** for this article is available online.

### Acknowledgements

We thank Dr. Eric Olson (UT Southwestern) for providing the MCK-miR-499 mice and Dr. Yong Liu (Wuhan University) for critical reading the manuscript. This work was supported by grants from the National Natural Science Foundation of China (31471110, 81400821), Ministry of Science and Technology of China (973 Program 2015CB856300), Natural Science Foundation of Jiangsu Province (BK20140600) to Z.G., and NIH grant RO1DK045416 to D.P.K. We thank Xiaolei Zhang (Translational Research Institute for Metabolism and Diabetes, Florida Hospital, FL), Lun Kuang, Yun Yan, Xiaoshuang Hou, Zhongnan Ma, and Wenxu He (Nanjing University) for technical assistance.

### Author contributions

JL and XL contributed equally to this work and performed most of the experiments with assistance from DZ, LLa, TF, LLi, LX, and YK, and QZ, RBV, M-SZ, DPK, and XG contributed reagents and provided scientific insight and discussion. ZG provided oversight of the study including experimental design and data interpretation and wrote the manuscript. All authors reviewed and contributed to the manuscript.

### Conflict of interest

The authors declare that they have no conflict of interest.

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
