## [Review Process File · EMBO Molecular Medicine]

Coupling of mitochondrial function and skeletal muscle fiber type by a miR-499/Fnip1/AMPK circuit

Jing Liu, Xijun Liang, Danxia Zhou, Ling Lai, Liwei Xiao, Lin Liu, Tingting Fu, Yan Kong, Qian Zhou, Rick B. Vega, Min-Sheng Zhu, Daniel P. Kelly, Xiang Gao and Zhenji Gan

Corresponding author: Zhenji Gan, Nanjing University

Review timeline:

Submission date:	06 March 2016
Editorial Decision:	07 April 2016
Revision received:	15 June 2016
Acceptance:	01 July 2016
Accepted:	01 July 2016

Transaction Report:

Editor: Roberto Buccione

1st Editorial Decision

07 April 2016

Thank you for the submission of your manuscript to EMBO Molecular Medicine. We have now received comments from the three Reviewers whom we asked to evaluate your manuscript

You will see that all three Reviewers provide thoughtful, detailed evaluations, are quite supportive of your work, and acknowledge its potential interest and importance. However, they also quite strikingly converge, albeit coming from different angles, on an fundamental issue that prevents us from considering publication at this time. I refer specifically to the missing causal link between Fnip1 and miR-499 in fibre metabolism and by extension the disconnection between the effects of miR-499, Fnip1 and PGC-1 α and the observations in the mdx mice.

The reviewers also mention additional, less critical concerns that would also need to be addressed.

In conclusion, while publication of the paper cannot be considered at this stage, we would be pleased to consider a suitably revised submission, provided that the Reviewers' concerns are addressed.

Please note that it is EMBO Molecular Medicine policy to allow a single round of revision only and that, therefore, acceptance or rejection of the manuscript will depend on the completeness of your responses, as outlined above and clearly addressing the crucial mechanistic and causal connection issues mentioned by the reviewers, in the next version of the manuscript.

As you know, EMBO Molecular Medicine has a "scooping protection" policy, whereby similar

findings that are published by others during review or revision are not a criterion for rejection. Although I clearly do not foresee such an instance in this case, I do ask you to get in touch with us after three months if you have not completed your revision, to update us on the status. Please also contact us as soon as possible if similar work is published elsewhere.

Please note that EMBO Molecular Medicine now requires a complete author checklist (<http://embomolmed.embopress.org/authorguide#editorial3>) to be submitted with all revised manuscripts. Provision of the author checklist is mandatory at revision stage; The checklist is designed to enhance and standardize reporting of key information in research papers and to support reanalysis and repetition of experiments by the community. The list covers key information for figure panels and captions and focuses on statistics, the reporting of reagents, animal models and human subject-derived data, as well as guidance to optimise data accessibility.

I also suggest that you carefully adhere to our guidelines for publication in your next version, including presentation of statistical analyses and our new requirements for supplemental data (see also below) to speed up the pre-acceptance process in case of a favourable outcome.

Finally, please note that we now mandate that all corresponding authors list an ORCID digital identifier. You may do so through our web platform upon submission and the procedure takes <90 seconds to complete. We also encourage the co-authors to supply an ORCID identifier, which will be linked to their name for unambiguous name identification.

I look forward to seeing a revised form of your manuscript as soon as possible.

***** Reviewer's comments *****

Referee #1 (Comments on Novelty/Model System):

The quality of the data is high and the appropriate mouse models and assays have been used. However, there appears to be a disconnect between the effect of miR-499, FNIP1 and PGC-1a (which does not influence fiber type) and the observed effect in the MDX mice, which involves an effect on fiber type.

Referee #1 (Remarks):

This manuscript describes the role of miR-499 in regulating skeletal muscle fiber type via a FNIP1-dependent mechanism. The authors identified PGC-1a among the upregulated genes in the miR-499 transgenic muscles, which appeared to be unrelated to muscle fiber type. This effect was due to a downregulation of FNIP1 and a subsequent effect on AMPK. Next the authors go on to show that reintroducing miR-499 in MDX mice activates the slow-oxidative muscle fiber type and improves muscle function.

This is well taken care of manuscript that shows a clear effect of miR-499 on muscle fiber type and FNIP1. However, it is unclear whether these 2 mechanisms are related. There are several key questions that need to be answered.

Comments:

- The authors use MCK-miR-499 transgenic animals to study the effect of miR-499 in skeletal muscle. This promoter drives the expression of miR-499 in all muscle types but induces a variable amount of overexpression in the different muscles. Is there an observable difference in effect between the different muscle types?
- In Figure 2 the authors show the upregulated genes. What about the downregulated genes? These should also be shown and indicated whether there are predicted targets for miR-499 among them.
- What does the change in LDHs mean in response to miR-499 overexpression?
- In Figure 3 the authors show that miR-499 overexpression leads to an increase in PGC-1a, which appears to be disconnected from muscle fiber type identity. Is this true for all muscle types?
- The authors show that FNIP1 is a direct target of miR-499 and responsible for the effect on PGC-1a via the phosphorylation of AMPK. Is the effect on PGC-1a blocked if AMPK is reduced?

- In Figure 6 the authors show a reduction in miR-499 in the MDX mice. However Figure 6c appears to show a slight increase in PGC-1 α in these mice. Can the authors explain this?
- Since miR-499 is downregulated in the MDX mice the authors should show an effect on FNIP1, pAMPK and PGC1 α and also whether this is reversed by miR-499 overexpression. Since there is an effect on fiber type, the observed protective effect in the MDX mice by overexpression of miR-499 might not be due to FNIP1 regulation.

Referee #2 (Comments on Novelty/Model System):

The authors attempt to show that the miR499 act through PGC1 α and Fnip1. To do so they used a PGC1-a KO murine model and miR499 Tg murine model, as well as a siRNA strategy for in vitro model of Fnip1 KD. The authors are suggesting that miR499 inhibits the expression of Fnip1 and thus up-regulates PGC1-a.

To reach such a conclusion, there is an important experiment missing. Indeed to confirm that miR499 has a key role the co-regulation of mitochondrial metabolism and fibre type, the authors should inhibit the expression of miR499 and confirm that PGC1 α is affected in such conditions, as well as the expression of Fnip1. In addition an inhibition of miR499 and Fnip1 should abolished the effect of each other. This type of experiments would increase the strength of the paper.

Referee #2 (Remarks):

Liu et al. describe a very interesting mechanism in the muscle field, linking the mitochondrial content and the fibre type. Based on the literature, they hypothesize that miR-499 and miR208b could be involved in the control of the mitochondrial biogenesis during fibre type switch. To test this hypothesis they studied the co-expression of miR499 and miR208b with Myh7 and Myh7b, and with the mitochondrial activity. Subsequently, all the paper is studying the role of miR499 on the mitochondrial metabolism. The authors attempt to show that the miR499 act through PGC1 α and Fnip1. To do so they used a PGC1-a KO murine model and miR499 Tg murine model, as well as a siRNA strategy for in vitro model of Fnip1 KD. The authors are suggesting that miR499 inhibits the expression of Fnip1 and thus up-regulates PGC1-a.

To reach such a conclusion, there is an important experiment missing. Indeed to confirm that miR499 has a key role the co-regulation of mitochondrial metabolism and fibre type, the authors should inhibit the expression of miR499 and confirm that PGC1 α is affected in such conditions, as well as the expression of Fnip1. In addition an inhibition of miR499 and Fnip1 should abolished the effect of each other. This type of experiments would increase the strength of the paper.

There are also some other points that need to be clarified, see below.

Points to clarify:

In the introduction the authors say that miR499 and miR208b might both be involved in the co-regulation of mitochondrial content and fibre type, and yet the authors just described the role of miR499.

- Figure 1A: The authors should show that the gene use to normalize the sample does not vary between sample. In addition, they should show that this gene is the best out of 5 for normalization (see ref Bustin et al., Clinical Chemistry 55:4 ; 611-622 (2009))
- Figure 1F: description of the lactate level measurements in blood samples is missing.
- Gene array: which stat were used? Anova, T-test ? I'm guessing the authors are meaning that they have analysed 3 independent samples per group. In addition, what Bio pathway means? Did the authors use only the Gene Ontology annotation? And if so how?
- Figure 2A: How big of the gene list was to upload to David Database? Was it a list of 100, 1000 genes? Was there another cut off than P<0.05 and 1.3 fold change used for this study?

- Figure 2G: the authors need to give a robust quantification of the percentage of MHC-I, as well as for the SDH staining and GPDH.
- Figure 3C: quantification of the cytochrome C level is missing
- Figure 6: miR499 compensate the mitophagy in mdx muscle and thus improve their muscle contractile capacity and frailty, results that corroborate with the literature.
- Supplemental figure 4: The authors should explain more clearly how they define a list of potential target for miR499: the Venn diagram suggests 10 genes, and yet 20 genes were tested by luciferase assay.

Referee #3 (Remarks):

The manuscript by Liu et al. describes the metabolic effects of miR-499 in skeletal muscle. miR-499 was already reported to drive type I myosin expression. Here, the authors demonstrate that miR-499 positively regulates muscle oxidative metabolism which plays a protective role in a dystrophic background. Mechanistically, miR-499 targets Fnip1, an AMPK interacting protein which regulates muscle fiber composition and mitochondrial function through PGC-1alpha. Overall the study is interesting and well executed. Of note, experiments were conducted exclusively in vivo or ex vivo. However, intriguing observations are sometimes not supported by compelling evidence. For example, a causal link between Fnip1 and miR-499 in determining fiber metabolism is missing. In addition, non-homogenous oxygen consumption rate measurements prevent comparisons between different mouse models and muscles.

Specific remarks:

OCR measurements are performed in myofibers, myotubes or isolated mitochondria in different experiments. However, the use of homogenous biological samples is compulsory in order to make comparisons between different models possible. For the same reason, a homogenous protocol of OCR measurements should be used throughout the study. Finally, OCR measurements should be performed in MCK-miR-499 versus NTG muscle samples.

Figure 2:

Analysis of MCK-miR-499 muscle metabolism should be completed.

mRNA expression data suggesting increased fatty acid uptake and glucose oxidation should be validated by quantitative analysis of these parameters.

Quantification of the number of ATPase, SDH and GPDH positive fibers should be performed in complete transversal muscle sections. Is mitochondrial activity due to increased mitochondrial number, as suggested by the increased PGC-1alpha activity? Mitochondrial DNA content should be measured.

Figure 3:

Additional experiments should be performed in order to determine to which extent PGC-1alpha deletion reverts miR-499 phenotype. Is OCR perturbed in 499Tg/PGC-1alpha KO muscles? Are fatty acid uptake and glucose oxidation impaired? Is mitochondrial DNA content reduced? Is ATPase, SDH and GPDH activity changed?

Figure 5:

The causal link between Fnip1 expression and miR-499 phenotype should be demonstrated. Overexpression of Fnip1 in MCK-miR499 muscles, which could be acutely achieved, should be performed and metabolic parameters (OCR, mitochondrial content and activity, fatty acids and glucose oxidation) should be measured.

In addition to Fnip1 mRNA analysis, Fnip1 protein expression should be measured in Fnip1 siRNA-transfected myotubes.

Figure 6:

Is Fnip1 involved in the protection exerted by miR-499 on mdx mice? Fnip1 expression should be measured in mdx/499Tg muscles.

In addition to increased number of central-nucleated fibers, also fiber size heterogeneity is augmented in mdx versus control animals. Is this parameter quantitatively reverted by miR-499 expression?

OCR should be measured in mdx/499Tg versus mdx samples. In addition, parameters indicative of

oxidative metabolism should be measured (as before, mitochondrial content and activity, fatty acids and glucose oxidation).

Mdx myofibers are especially susceptible to damage induced by eccentric contractions, such as those produced by downhill running. For this reason, exercise performance measurements upon repeated bouts of downhill running would be more informative.

1st Revision - authors' response

15 June 2016

Itemized Responses to Reviewer #1

“Referee #1 (Comments on Novelty/Model System):

The quality of the data is high and the appropriate mouse models and assays have been used. However, there appears to be a disconnect between the effect of miR-499, FNIP1 and PGC-1 α (which does not influence fiber type) and the observed effect in the MDX mice, which involves an effect on fiber type.

Referee #1 (Remarks):

This manuscript describes the role of miR-499 in regulating skeletal muscle fiber type via a FNIP1-dependent mechanism. The authors identified PGC-1 α among the upregulated genes in the miR-499 transgenic muscles, which appeared to be unrelated to muscle fiber type. This effect was due to a downregulation of FNIP1 and a subsequent effect on AMPK. Next the authors go on to show that reintroducing miR-499 in MDX mice activates the slow-oxidative muscle fiber type and improves muscle function.

This is well taken care of manuscript that shows a clear effect of miR-499 on muscle fiber type and FNIP1.

We wish to thank the Reviewer for a critical and constructive review. Our itemized responses to the issues raised are provided below:

“This is well taken care of manuscript that shows a clear effect of miR-499 on muscle fiber type and FNIP1.”

Thank you for the positive comment.

Major comments:

- 1. However, it is unclear whether these 2 mechanisms are related. There are several key questions that need to be answered.”*

The Reviewer raises the important question about the relative roles of miR-499 and Fnip1/PGC-1 α signaling in the observed rescue of the mdx phenotype. We submit that based on our results and that of the published literature that it is difficult, if not impossible, to sort out the relative contribution between these two key components of the regulatory cascade that has been unveiled in this work. First, we have demonstrated in the original Figure 6 that PGC-1 α is dramatically upregulated in mdx/miR-499Tg muscle. In addition, we have added new data demonstrating that Fnip1 protein levels are decreased in miR-499/mdx muscle (revised Fig. 6C). These results indicate a strong correlation, at the very least, between Fnip1/PGC-1 α signaling and miR-499 levels in the mdx model experiments. Secondly, it should be noted that there is significant published evidence that PGC-1 α and Fnip1 impact the mdx phenotype (work by others). Specifically, activation of PGC-1 α in skeletal muscle has been shown to improve muscular dystrophy (Handschin et al., *Genes Dev.* 2007; 21(7):770-83; Selsby et al., *PLoS One.* 2012;7(1):e30063; Chan et al., *Skelet Muscle.* 2014, 4(1):2). In addition, Fnip1 knockout mice on an mdx genetic background reduce the mdx phenotype (Reyes et al., *Proc Natl Acad Sci U S A.*

2015;112(2):424-9). Taken together, we conclude that whereas a shift in fiber type is also a key component of the “rescue” response, linkage to appropriate energetics is likely necessary. Indeed, this coordinate response is an important aspect of the message in this work, given that it could be an appealing target that coordinates multiple pathways towards reversing muscle diseases, including muscular dystrophy. These points have been clarified and re-emphasized in the revised Discussion (page 21, top paragraph).

2. *“The authors use MCK-miR-499 transgenic animals to study the effect of miR-499 in skeletal muscle. This promoter drives the expression of miR-499 in all muscle types but induces a variable amount of overexpression in the different muscles. Is there an observable difference in effect between the different muscle types?”*

Our data indicates that the miR-499-mediated regulation of oxidative fiber type is relevant to multiple muscle types based on the following: (1) we have added new data demonstrating that the slow-twitch contractile gene expression is increased in all soleus, gastrocnemius (GC), and white vastus (WV) muscles (the latter two are quite marked) in the MCK-miR-499 mice (revised Fig. EV2C); (2) as shown in original Supplemental Figure 1C, the protein levels of PGC-1a and myoglobin are also increased in multiple muscle types in the MCK-miR-499 mice (revised Fig. EV1C). Interestingly, the induction of the mitochondrial related program (such as PGC-1a and myoglobin) appears to be much greater in fast-twitch (WV and GC) muscle than in slow-twitch (soleus).

3. *“In Figure 2 the authors show the upregulated genes. What about the downregulated genes? These should also be shown indicated whether there are predicted targets for miR-499 among them.”*

The downregulated genes (a total of 2105 genes, Fold change < -1.2) were displayed in the venn diagram in original Supplemental Figure 4. As requested, we have now added a new table showing downregulated miR-499 targets, predicted by TargetScan and MicroCosm, in MCK-miR-499 muscle (Appendix Table S3). A total of 120 miR-499 mRNA targets predicted by Target-Scan and MicroCosm were depressed in MCK-miR-499 GC muscle, including the well-established miR-499 target, Sox6. Thus, miR-499 activation in skeletal muscle is associated with downregulation of putative muscle mRNA targets.

4. *“What does the change in LDHs mean in response to miR-499 overexpression?”*

We demonstrated previously that lactate dehydrogenase B (*Ldhb*), a glucose oxidation biomarker, is linked to mitochondrial oxidative capacity in mice (Gan et al, *Genes Dev.* 2011;25(24):2619-30). The *Ldhb/Ldha* gene expression ratio is increased in MCK-miR-499 muscle. These results were of interest because the *Ldhb* isoenzyme favors the reaction that converts lactate to pyruvate which, in turn, provides substrate for the mitochondrial TCA cycle (glucose oxidation), whereas *Ldha* favors the reverse reaction to produce lactate from pyruvate generated by glycolysis. Therefore, miR-499 drives an *LDH* isoenzyme shift that diverts pyruvate into the mitochondrion to fully catabolize glucose for maximal ATP production, contributing to enhanced exercise capacity and mitochondrial activity, and decreased post-exercise blood lactate in MCK-miR-499 mice. This point is further emphasized in the revised Discussion (page 20, top paragraph).

5. *“In Figure 3 the authors show that miR-499 overexpression leads to an increase in PGC-1a, which appears to be disconnected from muscle fiber type identity. Is this true for all muscle types?”*

We agree that there is a disconnect between miR-499 and PGC-1a on muscle fiber type identity as provided in the original manuscript Figure 3D and Supplemental Figure 3, and in the original text, page 10, top paragraph. Our current and published results indicate that

PGC-1 α , while capable of stimulating miR-499-mediated shifts in fiber type, is dispensable for this process. We found no decrease in MHC1 percentage in the GC muscle of 499Tg/PGC-1a mKO mice compared to MCK-miR-499 mice (original Fig. 3D). Similar observations were made in the soleus muscle of the 499Tg/PGC-1a mKO mice (original Supplemental Fig. 3). We have provided further experimental data demonstrating that the expression of slow-twitch muscle fiber genes was activated by miR-499 but not affected by disruption of PGC-1a in all soleus, GC, and WV muscle. These new results further support our conclusion that PGC-1a is dispensable for the miR-499-mediated induction in slow-twitch muscle fiber type in multiple muscle types across a range of fiber type proportions and oxidative capacity. The new data have been added to revised Fig. EV2C and in the text (page 10, bottom paragraph).

6. “The authors show that FNIP1 is a direct target of miR-499 and responsible for the effect on PGC-1a via the phosphorylation of AMPK. Is the effect on PGC-1a blocked if AMPK is reduced?”

Thank you for this query. To determine whether AMPK signaling is involved in this mechanism, the effect of AMPK inhibitor compound C was assessed together with the presence of Fnip1 siRNA-mediated knockdown. As predicted, we have found that AMPK inhibition abolished the Fnip1 siRNA-mediated induction of PGC-1a mRNA. These results further establish the importance of AMPK in this mechanism. These new data are included in a revised Fig. 5C and in the text (page 13, top paragraph).

7. “In Figure 6 the authors show a reduction in miR-499 in the MDX mice. However Figure 6c appears to show a slight increase in PGC-1a in these mice. Can the authors explain this?”

The Reviewer raises an important point. However, we DO NOT believe that PGC-1a protein levels are significantly increased in mdx muscle. There are several lines of evidence supporting this. First, it should be noted that PGC-1a protein levels are hardly

detectable in WT GC muscle. A PGC-1a Western Blot in mdx muscle (see Fig. 1A and 1B below) shows a slight increase in an upper non-specific band (see also original Fig. EV2A). Whereas no difference in PGC-1a protein levels was observed in the GC muscle of mdx mice compared to WT controls, probably due to very low basal PGC-1a levels, activation of miR-499 in mdx muscle resulted in dramatic increase in expression of PGC-1a. The quantification of PGC-1a Western Blot is included in the revised Fig. 6C. Indeed, the known PGC-1a targets, including *Cyts*, *Cox7a1*, *Ldhd*, are decreased in mdx muscle compared to WT control (see Fig. 1C below), which is consistent with previously reported findings that PGC-1a is decreased in mdx muscle (Matsakas et al., *FASEB J.* 2013;27(10):4004-16). We have replaced the original Figure 6C with the new PGC-1a gel to clarify this point.

8. *“Since miR-499 is downregulated in the MDX mice the authors should show an effect on FNIP1, pAMPK and PGC1a and also whether this is reversed by miR-499 overexpression. Since there is an effect on fiber type, the observed protective effect in the MDX mice by overexpression of miR-499 might not be due to FNIP1 regulation.”*

We agree that it is important to show the effect of miR-499 on Fnip1/PGC-1a signaling in the mdx model. We have provided the following points and additional experimental data as requested. First, we have added new data demonstrating that Fnip1 protein showed a *trend* towards *increase* in mdx muscle but significantly suppressed in mdx/miR-499Tg muscle. These important new data, consistent with our mechanistic model, have been added to the revised Fig. 6C and in the text (page 15, top paragraph). Second, we found an increase in p-AMPK levels in mdx muscle and this is reversed by miR-499 activation. The activation of p-AMPK is consistent with cellular sensing of relative energetic deficiency within mdx dystrophic myofibers and with previous reports (Pauly et al, *Am J Pathol.* 2012, 181(2):583-92). We speculate that the energetic deficiency state in mdx myofibers would trigger the ADP/ATP sensor AMPK signaling, and the correction of AMPK activity in mdx/miR-499Tg muscle could also reflect a restored energy production capacity. The new AMPK data have been added to the revised Fig. EV4A and in the text (page 15, top paragraph).

Finally, we believe that the coordinated regulation of both muscle fiber type and Fnip1/PGC-1a-dependent mitochondrial function by miR-499 ameliorates mdx muscular dystrophy. Our findings suggest that Fnip1/PGC-1a is highly related to the prevention of muscular dystrophy in mdx/miR-499Tg model. This point has been re-enforced and clarified in the revised Discussion (page 21, top paragraph).

Itemized Responses to Reviewer #2

Referee #2 (Comments on Novelty/Model System):

The authors attempt to show that the miR499 act through PGC1a and Fnip1. To do so they used a PGC1-a KO murine model and miR499 Tg murine model, as well as a siRNA strategy for in vitro model of Fnip1 KD. The authors are suggesting that miR499 inhibits the expression of Fnip1 and thus up-regulates PGC1-a.

To reach such a conclusion, there is an important experiment missing. Indeed to confirm that miR499 has a key role the co-regulation of mitochondrial metabolism and fibre type, the authors should inhibit the expression of miR499 and confirm that PGC1a is affected in such conditions, as well as the expression of Fnip1. In addition an inhibition of miR499 and Fnip1 should abolished the effect of each other. This type of experiments would increase the strength of the paper.

Referee #2 (Remarks):

Liu et al. describe a very interesting mechanism in the muscle field, linking the mitochondrial content and the fibre type. Based on the literature, they hypothesize that miR-499 and miR208b could be involved in the control of the mitochondrial biogenesis during fibre type switch. To test this hypothesis they studied the co-expression of miR499 and miR208b with Myh7 and Myh7b, and with the mitochondrial activity. Subsequently, all the paper is studying the role of miR499 on the mitochondrial metabolism. The authors attempt to show that the miR499 act through PGC1a and Fnip1. To do so they used a PGC1-a KO murine model and miR499 Tg murine model, as well as a siRNA strategy for in vitro model of Fnip1 KD. The authors are suggesting that miR499 inhibits the expression of Fnip1 and thus up-regulates PGC1-a."

We wish to thank the Reviewer for a critical and constructive review. Our itemized responses to the issues raised are provided below:

"Liu et al. describe a very interesting mechanism in the muscle field, linking the mitochondrial content and the fibre type."

Thank you for the positive comment.

Major comments:

- "To reach such a conclusion, there is an important experiment missing. Indeed to confirm that miR499 has a key role the co-regulation of mitochondrial metabolism and fibre type, the authors should inhibit the expression of miR499 and confirm that PGC1a is affected in such conditions, as well as the expression of Fnip1. In addition an inhibition of miR499 and Fnip1 should abolished the effect of each other. This type of experiments would increase the strength of the paper."*

The Reviewer has raised an interesting and important point about the causal link between the miR-499 and Fnip1. To assess the requisite role of miR-499 in the control of mitochondrial function in the absence of over-expression and determine whether Fnip1 is involved in this mechanism, miR-499 and Fnip1 loss-of-function studies were conducted in wild-type mouse primary skeletal myotubes. Because miR-499 and miR-208b function redundantly (see our response in Comment 2, below), we first conducted antisense-mediated inhibition of both miR-499 and miR-208b. As predicted, we have found that inhibition of both miRNAs modestly reduced PGC-1a mRNA levels in WT skeletal myotubes; conversely, Fnip1 mRNA levels are slightly but significantly increased following inhibition of both miRNAs. The observed increase in Fnip1 mRNA in inhibition of both miRNAs is consistent with miR-499 suppression of Fnip1. These results are now included in a revised Fig. 5F and in the text (page 13, bottom paragraph).

To assess the relevance of Fnip1 in miR-499-mediated mitochondrial function, the effect of miR-499/miR-208b inhibition on myocyte oxygen consumption rates (OCR) was assessed alone and together with the presence of Fnip1 siRNA. The results are interesting and consistent with the proposed mechanistic model. We have found that inhibition of both miRNAs resulted in a significant decrease in maximal OCR in the presence of the uncoupler FCCP, and Fnip1 knockdown prevented the miR-499/miR-208b inhibition-mediated repression of myocyte OCR. These results further establish the relevance of miR-499-mediated regulation of mitochondrial function and demonstrate the importance of Fnip1 inhibition in this mechanism. These important new data have been added to the revised Fig. 5G and in the text (page 13, bottom paragraph and page 14, top paragraph).

- "There are also some other points that need to be clarified, see below. Points to clarify: In the introduction the authors say that miR499 and miR208b might*

both be involved in the co-regulation of mitochondrial content and fibre type, and yet the authors just described the role of miR499.”

It has been established that miR-208b and miR-499 are functionally redundant, regulating many, if not all, of the same targets. Although effective loss-of-function would be predicted to require targeting of both miR-499 and miR-208b, one would predict that activation of either miRNA is capable of activating the muscle mitochondrial oxidative program. Our recent findings have suggested that miR-499 may be more dynamically regulated and, therefore, more important in humans (Gan et al, *J Clin Invest.* 2013;123(6):2564-75). As such, we mainly focused on miR-499 in this study.

3. “Figure 1A: The authors should show that the gene use to normalize the sample does not vary between sample. In addition, they should show that this gene is the best out of 5 for normalization (see ref Bustin et al., *Clinical Chemistry* 55:4 ; 611-622 (2009))”

As suggested, we have added more detail in regards to RT-qPCR analysis in the Methods section, including RNA quality and integrity assessment, reverse transcription, and qPCR protocol (page 27, top paragraph). Of note, we have determined the PCR efficiency of each individual gene primers by measuring serial dilutions of cDNA. For the reference gene, we have used 36B4 for quite a long time in our lab because we find it is stably expressed and least variable across many tissues and cell lines we tested. Regarding the RT-qPCR in Figure 1A, whereas we did see slight 36B4 Cq difference between myoblast and myotube (blast Cq: 22.9 ± 0.1 vs. myotube Cq: 24.4 ± 0.1), the 36B4 Cq difference between myoblast and myotube is relatively moderate compared to the huge induction of myosin *Myh7* and *Myh7b* gene expression. For the miRNA assay, sample normalization is carried out using snoRNA202 because this snoRNA shows the highest abundance and least variability across normal tissues and cell lines, and is one of the 2 best endogenous controls for mouse (Application Note Taqman MicroRNA Assays: Endogenous Controls for Real-Time Quantitation of miRNA using Taqman MicroRNA Assays). Similarly, compared to the huge induction of miR-499 and miR-208b, the snoRNA202 Cq difference between myoblast and myotube (blast Cq: 22.7 ± 0.1 vs. myotube Cq: 21.9 ± 0.1) is relatively moderate.

Figure 2. RT-qPCR analysis of *Myh7b* and *Myh7* levels during differentiation of myoblasts into mature myotubes using RPII as reference gene (n = 3). * $P < 0.05$ vs. corresponding controls. Values represent mean \pm SEM, and shown as arbitrary units (AU) normalized to myoblast controls.

The RNA polymerase II (RPII) gene has shown a strong correlation with the total amount of mRNA present in the sample, and has been shown to be the best choice (out of 13 commonly used) for a reference gene when using quantitative for RNA transcription analysis (Radonić et al., *Biochem Biophys Res Commun.* 2004;313(4):856-62). We have conducted additional RT-qPCR using RPII as reference gene. The RPII Cq show less difference between myoblast and myotube (blast Cq: 22.6 ± 0.2 vs. myotube Cq: 23.1 ± 0.2), and we see a similar huge induction of *Myh7* and *Myh7b* in myotubes compared to

myoblast (see Fig. 2), leading to the same conclusion. Therefore, we are confident in our

conclusion that the induced expression of *Myh7* and *Myh7b* genes paralleled the elevated expression of miR-208b and miR-499 during differentiation of skeletal myoblasts into myotubes.

4. *“Figure 1F: description of the lactate level measurements in blood samples is missing.”*

The lactate level measurements in blood samples were in the original Methods section under exercise studies (original page 21, line 11 and 12).

5. *“Gene array: which stat were used? Anova, T-test ? I'm guessing the authors are meaning that they have analysed 3 independent samples per group. In addition, what Bio pathway means? Did the authors use only the Gene Ontology annotation? And if so how?
- Figure 2A: How big of the gene list was to upload to David Database? Was it a list of 100, 1000 genes? Was there another cut off than $P < 0.05$ and 1.3 fold change used for this study?”*

The gene array data discussed in this manuscript have been deposited in NCBI's Gene Expression Omnibus and are accessible through GEO Series accession number GSE72581 accessible in the following link:

<http://www.ncbi.nlm.nih.gov/geo/query/acc.cgi?token=exovgaaadpwnzwl&acc=GSE72581>. Three independent samples per group were analyzed. Partek Analysis was used to identify genes differentially expressed. This information was provided in the original manuscript in the Methods section, original page 23, line 2 and 3.

In addition, we have added more details to the Methods section (Page 26, top paragraph). In brief, three independent samples per group were analyzed. The gene array data was analyzed using RMA algorithm using the Expression Console software Version 1.1 default settings. Partek Analysis using ANOVA was used to identify differentially expressed genes. For the DAVID pathway analysis (Nature Protocols 2009; 4(1):44 & Genome Biology 2003; 4(5):P3), a total of 1002 upregulated genes ($P < 0.05$ and 1.3 fold change) were uploaded into DAVID Bioinformatics Resources 6.7. The Functional Annotation analysis was used to interpret data, and the pathways were reviewed using the cellular compartments defined by Gene_Ontology, which is ranked by p value of a regulated term in Figure 2A.

6. *“Figure 2G: the authors need to give a robust quantification of the percentage of MHC-I, as well as for the SDH staining and GPDH.”*

The quantification of the staining in gastrocnemius (GC) muscle has been a challenge, in part, due to the heterogeneity of this muscle. For the SDH and GPDH staining, we have quantified numerous transverse muscle sections and present this as percentage of positive staining myofibers. The average percentage of the SDH positive staining was significantly increased in GC muscle of MCK-miR-499 mice (NTG, $30.5 \pm 2.0\%$ vs. MCK-miR-499, $58.7 \pm 1.3\%$, $n = 4$ mice per group, $p < 0.0001$), while the average percentage of the GPDH positive staining was significantly decreased in GC muscle of MCK-miR-499 mice (NTG, $74.8 \pm 1.5\%$ vs. MCK-miR-499, $31.6 \pm 1.4\%$, $n = 5$ mice per group, $p < 0.0001$). For the ATPase staining, we have quantified numerous sections and present this as number of type I fibers per section (NTG, 126 ± 42 per section vs. MCK-miR-499, 551 ± 43 per section, $n = 5$ mice per group, $p = 0.0006$). This information has been added to the Results section (page 8, bottom paragraph and page 9, top paragraph).

7. *“Figure 3C: quantification of the cytochrome C level is missing”*

As requested, we have added the quantification of the cytochrome C level to the revised Fig. 3C.

8. *“Figure 6: miR499 compensate the mitophagy in mdx muscle and thus improve their muscle contractile capacity and frailty, results that corroborate with the literature.”*

We thank the Reviewer for this point and have provided a broader discussion and referencing of mitophagy in mdx muscle in the revised manuscript (page 21, top paragraph). See new reference (Pauly et al, *Am J Pathol.* 2012, 181(2):583-92).

9. *“Supplemental figure 4: The authors should explain more clearly how they define a list of potential target for miR499: the Venn diagram suggests 10 genes, and yet 20 genes were tested by luciferase assay.”*

The TargetScan and MicroCosm programs were used to identify putative target mRNA for miR-499, and this list was cross-matched for genes that are downregulated in MCK-miR-499 muscle, which is diagrammed in the original Figure 4. We have now added a new table showing the full list of 120 downregulated miR-499 targets, predicted by TargetScan and MicroCosm, in MCK-miR-499 muscle. Of note, it has been recommended that combining the results of different target prediction programs to look for overlap in predicted miRNAs targets between the different programs will result in the highest specificity but lowest sensitivity. In order to maintain both high sensitivity and specificity, we also chose those putative targets that are known to be involved in the regulation of energy metabolism or mitochondrial function, in addition to the overlapping putative targets. Thus, in the end, we tested 20 putative targets by in vitro UTR luciferase assay. These points have been added to the Expanded View Figure legend (page 53, bottom paragraph).

Itemized Responses to Reviewer #3

“Referee #3 (Remarks):

The manuscript by Liu et al. describes the metabolic effects of miR-499 in skeletal muscle. miR-499 was already reported to drive type I myosin expression. Here, the authors demonstrate that miR-499 positively regulates muscle oxidative metabolism which plays a protective role in a dystrophic background. Mechanistically, miR-499 targets Fnip1, an AMPK interacting protein which regulates muscle fiber composition and mitochondrial function through PGC-1alpha.

Overall the study is interesting and well executed. Of note, experiments were conducted exclusively in vivo or ex vivo. However, intriguing observations are sometimes not supported by compelling evidence. For example, a causal link between Fnip1 and miR-499 in determining fiber metabolism is missing. In addition, non-homogenous oxygen consumption rate measurements prevent comparisons between different mouse models and muscles.

We wish to thank the Reviewer for a critical and constructive review. Our itemized responses to the issues raised are provided below:

“Overall the study is interesting and well executed. Of note, experiments were conducted exclusively in vivo or ex vivo.”

We appreciate this point.

Major comments:

1. *“Specific remarks:*

OCR measurements are performed in myofibers, myotubes or isolated mitochondria in different experiments. However, the use of homogenous biological samples is compulsory in order to make comparisons between different models possible. For the

same reason, a homogenous protocol of OCR measurements should be used throughout the study. Finally, OCR measurements should be performed in MCK-miR-499 versus NTG muscle samples.”

The Reviewer has raised an important issue about using different mitochondrial functional readouts, including the Seahorse-based isolated mitochondrial and O2K-based permeabilized myofibers OCR measurements in this study. We agree that the OCR results are not exactly the same but lead to the same conclusion. It should be noted, as described following: First, it has been shown that respiratory results using permeabilized tissues are in excellent agreement with data on isolated mitochondria based on human muscle studies (Gnaiger, *Int J Biochem Cell Biol.* 2009;41(10):1837-45.; Rasmussen et al., *Am J Physiol Endocrinol Metab.* 2001;280(2):E301-7). Second, the interpretation is very different comparing changes in respiratory flux per mass of tissue, or per mitochondrial protein. For example, soleus muscle has more mitochondrial content compared to white vastus (WV) muscle, and it has been known that mdx muscle have less mitochondrial content compared to WT controls. Therefore, we conducted Seahorse-based isolated mitochondrial OCR measurement and normalized data per mitochondrial protein in Fig. 1D and Fig. 6B. While we took advantage of O2K-based permeabilized muscle OCR in Fig. 1G because we found no change in mitochondrial DNA content in NTG and MCK-miR-499Tg muscle (Fig. EV1D).

As shown in original Figure 1G, we have performed O2K-based permeabilized myofibers OCR measurements in the muscle of the MCK-miR-499 mice and corresponding NTG controls. Mitochondrial respiration rates were significantly higher in MCK-miR-499 muscle compared to the NTG controls (original page 7, top paragraph).

2. *“Figure 2: Analysis of MCK-miR-499 muscle metabolism should be completed. mRNA expression data suggesting increased fatty acid uptake and glucose oxidation should be validated by quantitative analysis of these parameters. Quantification of the number of ATPase, SDH and GPDH positive fibers should be performed in complete transversal muscle sections. Is mitochondrial activity due to increased mitochondrial number, as suggested by the increased PGC-1alpha activity? Mitochondrial DNA content should be measured.”*

We thank the Reviewer and provide the following points and additional experimental data. We have found that miR-499 drives a broad array of mitochondrial oxidative programs. We primarily focused on the mitochondrial respiration function in this study, and we believe that the fuel specific utilization is beyond the scope of the current study, but would be an important focus of study for the future.

The quantification of the staining in MCK-miR-499 muscle has been added to the Results section (page 8, bottom paragraph and page 9, top paragraph). In brief, we have quantified numerous transverse muscle sections and present as percentage of positive staining myofibers for SDH and GPDH staining; For ATPase staining, we have quantified and present this as number of type I fibers per section.

As suggested, we have added new data showing no difference in mtDNA content in muscle of MCK-miR-499 mice compared to NTG controls. These results suggest that PGC-1a effects mitochondrial respiration capacity without changes in mitochondrial content in MCK-miR-499 muscle, which is consistent with previous report (Rowe et al., *Cell Rep.* 2013;3(5):1449-56). These results have been added to the revised Fig. EV1D and in the text (page 9, bottom paragraph and page 10, top paragraph).

3. *“Figure 3: Additional experiments should be performed in order to determine to which extent PGC-1alpha deletion reverts miR-499 phenotype. Is OCR perturbed in 499Tg/PGC-1alpha KO muscles? Are fatty acid uptake and glucose oxidation impaired? Is mitochondrial DNA content reduced? Is ATPase, SDH and GPDH activity changed?”*

As suggested, we have conducted additional physiological and metabolic phenotyping of the 499Tg/PGC-1a KO mouse lines to characterize the effect of PGC-1a on miR-499Tg phenotype. As expected, disruption of PGC-1a abolished the miR-499-mediated enhancement of exercise capacity and the RER lowering effect during exercise. In addition, muscle-specific disruption of PGC-1a prevented the miR-499-mediated reduction of blood lactate levels following exercise. Together, these results provide further evidence that PGC-1a is required for the miR-499-mediated increase in oxidative metabolism in muscle. These results have been added to revised Fig. 3E and F and addressed in the Text (page 11, second paragraph).

We did not pursue mitochondrial DNA measurement in 499Tg/PGC-1a mKO muscle because we did not find mitochondrial DNA content to be changed by miR-499 (Fig. EV1D). The average percentage of the SDH positive staining was significantly decreased in GC muscle of 499Tg/PGC-1a mKO mice compared to 499Tg. Interestingly, however, no change in GPDH staining and MHC1 immunofluorescence was observed in the GC muscle of 499Tg/PGC-1a mKO mice compared to MCK-miR-499 mice. The quantification information has been added to the Results section (page 10, bottom paragraph).

4. *“Figure 5:*

The causal link between Fnip1 expression and miR-499 phenotype should be demonstrated. Overexpression of Fnip1 in MCK-miR499 muscles, which could be acutely achieved, should be performed and metabolic parameters (OCR, mitochondrial content and activity, fatty acids and glucose oxidation) should be measured. In addition to Fnip1 mRNA analysis, Fnip1 protein expression should be measured in Fnip1 siRNA-transfected myotubes.”

The Reviewer has raised an interesting and important point about the causal link between the miR-499 and Fnip1. Please also see our response to Comment 1 of Reviewer #2. Indeed, we have conducted *in vivo* Fnip1 plasmid muscle electroporation experiment, but we were unable to assess Fnip1 effect in MCK-miR-499 muscle due to technical reasons. However, we have conducted miR-499 and Fnip1 loss-of-function studies in wild-type mouse primary skeletal myotubes to address this question. Because miR-499 and miR-208b have redundant functions, we first conducted antisense-mediated inhibition of both miR-499 and miR-208b. We have found that inhibition of both miRNAs modestly reduced PGC-1a mRNA levels and, conversely, induced Fnip1 mRNA levels. The effect of miR-499/miR-208b inhibition on myocyte oxygen consumption rates (OCR) was next assessed alone and together with the presence of Fnip1 siRNA. The results are interesting and consistent with the proposed mechanistic model. We have found that inhibition of both miRNAs resulted in significant decrease in maximal OCR in the presence of the uncoupler FCCP, and Fnip1 knockdown prevented the miR-499/miR-208b inhibition-mediated repression of myocyte OCR. These results further establish the relevance of miR-499-mediated regulation of mitochondrial function and demonstrate the importance of Fnip1 inhibition in this mechanism. These important new data have been added to the revised Fig. 5F and G and in the text (page 13, bottom paragraph and page 14, top paragraph).

In addition, we have also conducted studies to assess the levels of Fnip1 protein in Fnip1 siRNA-transfected myotubes (see Fig. 3, below). Whereas the Fnip1 antibody detects a ~130 kD protein band specifically reduced in Fnip1 siRNA-transfected myotubes, we think that the data is not conclusive, so we hesitate to include these results in the revised manuscript.

Figure 3. Representative western blot analysis performed on extracts of myotubes subjected

to *Fnip1* siRNA or control (Con) siRNA using *Fnip1* antibodies.

5. *“Figure 6:
Is Fnip1 involved in the protection exerted by miR-499 on mdx mice? Fnip1 expression should be measured in mdx/499Tg muscles.”*

Thank you for this query. We have added new data demonstrating that *Fnip1* protein showed a trend towards increase in mdx muscle, but significantly suppressed in mdx/miR-499Tg muscle, which is consistent with our mechanistic model. These important new data are provided in a revised Fig. 6C and in the text (page 15, top paragraph). In addition, *Fnip1* knockout mice in an mdx genetic background have recently been shown to be protected against muscular dystrophy (Reyes et al., *Proc Natl Acad Sci U S A.* 2015;112(2):424-9), consistent with the protective effect of miR-499 on mdx mice. The observed strong correlation between *Fnip1* and miR-499 levels in mdx model suggests that *Fnip1*/PGC-1 α signaling is highly related to the prevention of muscular dystrophy in the mdx/miR-499Tg model.

6. *“In addition to increased number of central-nucleated fibers, also fiber size heterogeneity is augmented in mdx versus control animals. Is this parameter quantitatively reverted by miR-499 expression?”*

Thank you for this query. As suggested, we have quantified the fiber size distribution from numerous cross-sections of the GC muscle. Consistent with previous studies, mdx mice had a higher variation in fiber size and a higher number of small fibers compared to WT mice. The mdx/miR-499Tg mice had a lower variation in fiber size, and had fewer small fibers compared to mdx alone. This information has been included in a revised Fig. EV5A and in the text (page 16, second paragraph).

7. *“OCR should be measured in mdx/499Tg versus mdx samples. In addition, parameters indicative of oxidative metabolism should be measured (as before, mitochondrial content and activity, fatty acids and glucose oxidation).”*

We did not include such studies in the original manuscript given the large amount of metabolic phenotyping data from mdx/miR-499Tg mice already included. However, as requested, we have conducted additional mitochondrial content and OCR activity measurements. We have found that, as predicted, mitochondrial respiration capacity was significantly increased in mdx/miR-499Tg muscle compared to mdx mice. Interestingly, the mitochondrial DNA content remains unchanged in mdx/miR-499Tg muscle compared to mdx mice, consistent with no changes in mitochondrial content in MCK-miR-499 muscle. These important new data further support the conclusion that activation of miR-499 restored the mitochondrial bioenergetics capacity in mdx mice and have been added to the revised Fig. EV4D and E, and in the text (page 15).

8. *“Mdx myofibers are especially susceptible to damage induced by eccentric contractions, such as those produced by downhill running. For this reason, exercise performance measurements upon repeated bouts of downhill running would be more informative.”*

This is an excellent question. We are interested in this, as we have found that exercise performance (Run-to exhaustion) in mdx mice is significantly rescued by miR-499 activation (original Fig. 6F and G). However, we feel that these experiments are outside the scope of this manuscript, but would be an important set of future experiments.

Acceptance

01 July 2016

Please find enclosed the final reports on your manuscript. We are pleased to inform you that your manuscript is accepted for publication and is being sent to our publisher to be included in the next available issue of EMBO Molecular Medicine.

Congratulations on your interesting work,

***** Reviewer's comments *****

Referee #1 (Remarks):

The authors were able to improve the manuscript by adding in the majority of the data that were requested by the reviewers. There are no further comments

Referee #2 (Remarks):

The authors have improved their paper and answered very well to all my concerns. Now the paper is clearly showing a link between miR499, PGC1a and Fnip1. In addition all statistics and bio-informatic analysis have been clarified.

Referee #3 (Remarks):

The authors have responded adequately to my queries

Corresponding Author Name: Zhenji Gan

Manuscript Number: EMM-2016-06372